# SoftMatcha 2: A Fast and Soft Pattern Matcher for Trillion-Scale Corpora

**Masataka Yoneda** [1 2]  **Yusuke Matsushita** [3]  **Go Kamoda** [4 5]  **Kohei Suenaga** [3 2]
**Takuya Akiba** [6 7]  **Masaki Waga** [3 2]  **Sho Yokoi** [5 7 8]

## Abstract

We present SoftMatcha 2, an ultra-fast and flexible search algorithm that enables search over trillion-scale natural language corpora in under 0.3 seconds while allowing *semantic* variations in the form of substitution, insertion, and deletion. Our approach employs string matching based on suffix arrays that scales well with corpus size, and represents words as vectors, which underpin its semantic flexibility. To mitigate the combinatorial explosion induced by the semantic relaxation of queries, our method is built on two key algorithmic ideas: dynamic corpus-aware pruning and fast exact lookup enabled by a disk-aware design. We theoretically analyze the efficiency of the proposed method, indicating that it can mitigate exponential growth in the search space. Empirically, on FineWeb-Edu (Lozhkov et al., 2024) (1.4T tokens), it achieves substantially lower search latency than existing methods: infini-gram (Liu et al., 2024), infini-gram mini (Xu et al., 2025), and SoftMatcha (Deguchi et al., 2025). As a practical application, our method uncovers benchmark contamination in training corpora that existing approaches miss, and it also benefits information retrieval and paraphrase detection. We also provide an online demo of fast, soft search across corpora in seven languages.[1,2]

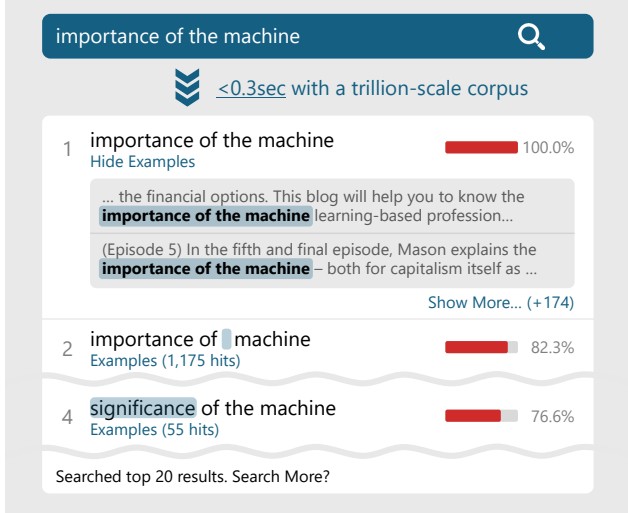

*Figure 1.* An example of search in SoftMatcha 2, which performs a soft search for trillion-scale corpora within 0.3 seconds, including word substitution, insertion, and deletion.

## 1. Introduction

Large language models (Meta, 2025; Gemma Team & Google Deepmind, 2025; OpenAI, 2025) owe their success to *massive corpora* (Penedo et al., 2024; Lozhkov et al., 2024; Soldaini et al., 2024), along with breakthroughs in statistical machine learning—including deep neural networks (Fukushima, 1980; Hinton et al., 2006; Vaswani et al., 2017), gradient descent-based optimization (Rumelhart et al., 1986; Bottou, 2010; Kingma & Ba, 2014), and self-supervised learning (Bengio et al., 2013; Gutmann & Hyvärinen, 2012). The significance of corpora in training language models has driven the demand for tools capable of quickly searching large corpora, e.g., to investigate the causes of unintended model behavior (Sun et al., 2025; Akyurek et al., 2022) and to estimate the degree of contamination in benchmark data (Deng et al., 2023; Xu et al., 2024).

Searching a corpus, or a collection of documents, has long been a topic of research in fields such as information retrieval. Modern corpus searching, however, poses distinct challenges: (1) The pretraining corpora for language models have grown rapidly **in scale**, reaching trillions of tokens (Ta-

---

[1]The University of Tokyo, Tokyo, Japan [2]National Institute of Informatics, Tokyo, Japan [3]Kyoto University, Kyoto, Japan [4]The Graduate University for Advanced Studies (SOKENDAI), Tokyo, Japan [5]National Institute for Japanese Language and Linguistics (NINJAL), Tokyo, Japan [6]Sakana AI, Tokyo, Japan [7]Tohoku University, Sendai, Japan [8]RIKEN, Tokyo, Japan. Correspondence to: Takuya Akiba <takiba@sakana.ai>, Masaki Waga <mwaga@fos.kuis.kyoto-u.ac.jp>, Sho Yokoi <yokoi@ninjal.ac.jp>.

*Proceedings of the $43^{rd}$ International Conference on Machine Learning*, Seoul, South Korea. PMLR 306, 2026. Copyright 2026 by the author(s).

[1] 🌐 **Project Page & Web Interface**:
https://softmatcha.github.io/v2/
[2] 🐙 **Source Code**: https://github.com/softmatcha/softmatcha2

*Table 1.* The growth of text corpus sizes over time.

| Corpus | Size |
|---|---|
| Brown Corpus (Kučera & Francis, 1967) | 1**M** tokens |
| Gigaword (Parker et al., 2011) | 4**B** tokens |
| FineWeb-Edu (Lozhkov et al., 2024) | 1.4**T** tokens |

ble 1), so search methods must scale to this size; (2) Because our language contains diverse paraphrases and orthographic variations, search must go beyond exact matching and retrieve text **similar** to the query; (3) Because queries are natural language, as in applications such as data contamination detection and information retrieval, their **word order** carries meaning and must be respected. Indeed, several recently developed corpus search tools (Liu et al., 2024; Xu et al., 2025; Deguchi et al., 2025) address some of these needs, but none addresses all three (Table 2).

We propose a fast full-text search algorithm that meets the three desiderata (1–3) mentioned above. To this end, we adopt a suffix-array-based approach that preserves **word order**, with exact-match lookup that **scales** to trillions of tokens. The challenge is that combining suffix arrays with word vectors, which enumerate candidate patterns **similar** to the query, yields a set that grows exponentially with query length. To keep this cost manageable, our algorithm combines two ideas: (i) *dynamic corpus-aware pruning* that eliminates unnecessary candidate patterns to curb this blowup (§ 3.3), and (ii) *disk-aware, staged suffix arrays*, requiring a single random disk access per exact-match lookup (§ 3.4). Our contributions are summarized as follows:

- We propose SoftMatcha 2, a fast full-text soft (or lexical-semantic) search algorithm for trillion-token-scale corpora that handles semantic similarity and token-count differences between queries and results, combining dynamic corpus-aware pruning and disk-aware, staged suffix arrays. (§ 3)
- We theoretically show that the proposed method mitigates the exponential growth in the search space with respect to the query length, leveraging statistical properties of natural language. (§ 4)
- We experimentally show that SoftMatcha 2 significantly outperforms existing methods. The 95th percentile latencies are $278.17\,\text{ms}$ for soft search and $0.34\,\text{ms}$ for exact search on FineWeb-Edu (Lozhkov et al., 2024) (1.4T tokens); the 95th percentile latencies for soft search are less than $400\,\text{ms}$ in Chinese (38.3B tokens) and Japanese corpora (169B tokens). (§ 5)
- We demonstrate the usefulness of SoftMatcha 2 in (1) information retrieval, (2) paraphrase detection, and (3) detecting contamination in the training corpora by questions in evaluation data. (§ 6)
- We release an online demo running on a 100B-token-scale English corpus.[1] Figure 1 illustrates an example.

## 2. Text Search at Scale

### 2.1. Existing Tools and Their Limitations

For searching trillion-scale corpora, **infini-gram** (Liu et al., 2024) and **infini-gram mini** (Xu et al., 2025) apply suffix arrays (Manber & Myers, 1990) and a variant of a suffix array (i.e., FM-indexes (Ferragina & Manzini, 2005)), which are powerful data structures for exact string matching, respectively. Our work follows this line, significantly improving exact-match search speed (§ 3.4 and § 5.3), while also incorporating semantic substitutions, insertions, and deletions. **SoftMatcha** (Deguchi et al., 2025) is, to our knowledge, the only prior work that successfully incorporates semantic elements into text search for large-scale corpora. Note that, due to the characteristics of the inverted index (Zobel & Moffat, 2006) they adopted, it does not scale well to trillion-scale corpora. We examine the differences between our work and these three prior studies in the experimental section.

Here, we have focused on directly related work—recent corpus search at scale for large language models. A much broader literature exists, however, spanning information retrieval, natural language processing, and beyond. For historical developments in corpus linguistics and natural language processing, as well as orthogonal approaches such as TF-IDF, BM25, and dense vector search, see § A.1.

### 2.2. Problem Formulation: Soft Text Search

**Notation and Inputs/Outputs**   We write $\mathcal{V}$ for the set of vocabulary tokens, $\mathcal{V}^*$ for the set of vocabulary token sequences (i.e., sequences over $\mathcal{V}$), and $\mathcal{C}$ for the corpus being searched. Given a *query* $\bar{q} = q_1 q_2 \cdots q_m \in \mathcal{V}^*$, a corpus $\mathcal{C}$, and a similarity threshold $\alpha > 0$, our algorithm returns a set of patterns $\bar{p} = p_1 p_2 \cdots p_n \in \mathcal{V}^*$ that are semantically similar to the query $\bar{q}$ with similarity $\alpha$ or more. We assume a similarity function $\text{sim}(\bar{q}, \bar{p}) \geq 0$ that scores how similar a query and a pattern are, computed from word (or token) embeddings; larger values indicate greater similarity.

**Indexing and Matching**   Following common practice in high-performance search, our algorithm proceeds in two stages. First, it constructs a query-independent *index* from the corpus. Second, at runtime, it uses this index for efficient soft pattern *matching*. This prior indexing is standard, and all baselines rely on it as well.

## 3. SoftMatcha 2: Soft and Fast Corpus Search

### 3.1. Building Blocks: Suffix Arrays and Word Vectors

We propose an algorithm for soft pattern matching (i.e., retrieving texts similar to the query, preserving the word order in the query), scalable to trillion-scale corpora. Our algo-

*Table 2.* Overview of corpus search methods, covering closely and indirectly related approaches to clarify the position of our method. [†]: ✓ indicates that results for an approximately 10-token query can be retrieved from one trillion tokens within one second.

| Paradigm | Algorithm | Example | Scalability | | Semantic Similarity | | Word Order | Exhaustive |
|---|---|---|---|---|---|---|---|---|
| | | | Indexing | Fast Search[†] | Lexical Similarity | Insertion & Deletion | | |
| Exact string matching | KMP / BM / Aho–Corasick | fgrep (Kernighan & M. Douglas, 1979) | ✗ | ✗ | ✗ | ✗ | ✓ | ✓ |
| | suffix array | infini-gram (Liu et al., 2024) | ✓ | ✓ | ✗ | ✗ | ✓ | ✓ |
| | FM-index | infini-gram mini (Xu et al., 2025) | ✓ | ✓ | ✗ | ✗ | ✓ | ✓ |
| Approximate string matching | DP × bit-parallelism | agrep (Wu & Manber, 1992) | ✗ | ✗ | ✗ | ✓ | ✓ | ✓ |
| Soft (lexical-semantic) string matching | inverted index × word vectors | SoftMatcha (Deguchi et al., 2025) | ✓ | ✗ | ✓ | ✗ | ✓ | ✓ |
| | suffix array × word vectors | **SoftMatcha 2** (Ours) | ✓ | ✓ | ✓ | ✓ | ✓ | ✓ |
| Regular expression matching | DFA / NFA / Virtual Machine | grep (Kernighan & M. Douglas, 1979) | ✗ | ✗ | ✗ | ✓ | ✓ | ✓ |
| Lexical similarity search | TF-IDF / BM25 | Elastic-search (Gormley & Tong, 2015) | ✓ | ✓ | ✗ | ✓ | ✗ | ✗ |
| Dense vector search | ANN × text vectors | FAISS (Douze et al., 2025) | ✓ | ✓ | coarse topic similarity | | ✗ | – |

rithm **scales** to trillion-scale corpora, preserving the **word order** in the query because an exact lookup based on a *suffix array* runs in logarithmic time in corpus size. *Word vectors* are also used in our direct predecessor SoftMatcha (Deguchi et al., 2025) to search for text **similar** to the query. Note that the inverted index runs in linear time in corpus size, and its scalability is limited. For related work on algorithms and data structures for string search, including suffix arrays and inverted indexes, see §A.2.

### 3.2. Challenges and Solutions

Unlike an inverted index, which supports "OR queries" that match any of several patterns at once (Deguchi et al., 2025), a suffix array offers fast lookup only for a single query at a time. Therefore, a naive approach to soft search with suffix arrays proceeds in two steps: given a query $\bar{q}$, it enumerates the patterns $w$ whose similarity to $\bar{q}$ is greater than or equal to $\alpha$, and then searches for an occurrence of each such $w$ using a suffix array. This approach faces two challenges: (i) the number of patterns $w$ increases exponentially with the query length; and (ii) lookup in a naive suffix array is slow due to many random disk accesses.

We address (i) by pruning the search space by dynamically exploiting the power-law distribution of $n$-grams in the corpus (§3.3). We solve (ii) by speeding up exact lookups using a new *disk-aware suffix array* exploiting the fact that the query length is typically short (§3.4). The combination of

**Algorithm 1** Obtain all patterns with similarity $\alpha$ or more that occur in the corpus (simplified).

$R_0 \leftarrow \{\text{""}\}$
**for** $i = 1$ **to** $m$ **do**
    $R'_i \leftarrow \varnothing$
    **for** $w \in R_{i-1}$ **do**
        $W \leftarrow \{w' \in \mathcal{V}^* \mid \mathsf{sim}(q_1 q_2 \cdots q_i, w \cdot w') \geq \alpha\}$
        $R'_i \leftarrow R'_i \cup \{w \cdot w' \mid w' \in W\}$
    **end for**
    $R_i \leftarrow \{w \in R'_i \mid w \text{ occurs in } \mathcal{C}\}$
**end for**
**return** $R_m$

the two gives a blazingly fast soft search for large corpora (see §5 for the experimental results). We also define our proposed similarity function later in §3.5, where word vectors enable soft substitutions, insertions, and deletions.

### 3.3. Dynamic Corpus-Aware Pruning of Search Space

To reduce the exponential growth of the search space in soft pattern matching, we employ **iterative pruning**, which drastically reduces the number of candidate patterns by exploiting the power-law distribution of $n$-grams in the corpus. Below, we describe our algorithm (shown in Algorithm 1), through an example of searching for "olympics gold medal" with a given initial value of $\alpha$ (see Figure 2 for illustration).

**Step 1.** Enumerate all the patterns $R'_1$ whose similarity to the first query token, "olympics", is at least $\alpha$. This step

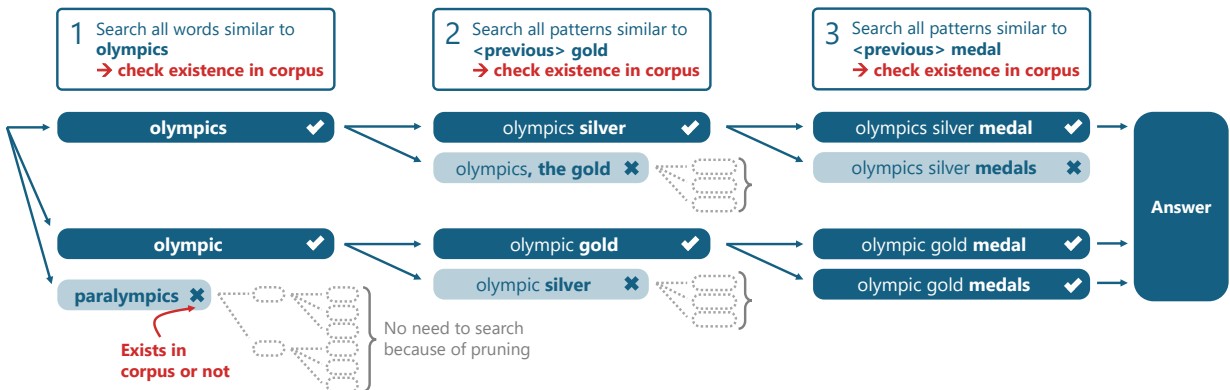

*Figure 2.* A sketch of our soft searching algorithm, when the query is "olympics gold medal". Without iterative pruning, we must search the gray-striped zone as well as the blue zone.

works only using the dictionary, without referencing the corpus. Then, filter $R_1'$ into the patterns $R_1$ that occur in the corpus, using exact lookup by the fast suffix array (§3.4).
**Step 2.** For each $w \in R_1$, enumerate all the patterns $w \cdot w'$ that have $w$ as a prefix and whose similarity to "olympics gold" is at least $\alpha$. We denote this set by $R_2'$. As before, filter $R_2'$ into those that appear in the corpus $R_2$.
**Step 3.** Compute $R_3'$ and $R_3$ analogously using the similarity to "olympics gold medal". The final result is $R_3$.

**Other Techniques.** To further improve performance, we also introduce the following two techniques. Details are shown in Appendix K. (i) $k$-**gram pruning**: For the 2-grams and 3-grams consisting only of high-frequency words, we check if they appear in the corpus beforehand and keep this information in RAM; at runtime, we determine if they appear without any extra disk access. (ii) **Last-bits pruning**: When the pattern $w \in R_{i-1}$ has only a few occurrences in the corpus, we can directly enumerate all its occurrences and check if they match the query, instead of appending some suffixes to $w$ and looking for them in the corpus. We use this pruning when $w$ has less than or equal to 50 occurrences.

### 3.4. Fast Disk-Aware Suffix Array

Our soft search method is built on top of exact corpus lookup, and its efficiency is essential for scaling our method to a trillion-scale corpus. A naive corpus lookup implementation based on a suffix array faces a critical bottleneck in index access: the index for a trillion-scale corpus is enormous ($> 5\,\text{TB}$), which prevents storing it entirely in RAM and requires storing it on disk (e.g., SSD). Since disk access latency is significantly slower ($\approx \times 10^3$) than RAM, reducing the number of disk accesses is vital.

To achieve this key goal, we newly design a **disk-aware, staged suffix array** that requires only *one* disk access per exact match query, while a standard suffix array requires $O(\log|\mathcal{C}|)$ accesses, where $|\mathcal{C}|$ is the corpus size. See Ap-

pendix L for the details. Our approach uses a two-staged indexing scheme, similar to Google's BigTable (Chang et al., 2008) in the high-level approach[3]: (1) identifying the range of potential matches in the first stage and (2) determining the precise match in the second stage, focusing on the identified area. For the maximum query length $L$, we first construct a sorted array $X$ of all contiguous $L$ tokens in the corpus. Then we construct a sparse array $Y = [X_0, X_B, X_{2B}, \ldots]$, where $X_i$ is the $i$-th element of the array $X$ and $B$ is a constant. This sparse array is stored in RAM. When we search for the position of an occurrence of query $\bar{q}$, we first find the approximate position (i.e., an index $i$ that $\bar{q}$ occurs in $Z = [X_{iB}, X_{iB+1}, \ldots, X_{(i+1)B-1}]$ if it exists in $X$) using $Y$, and then find the exact position in $X$ using $Z$. If $B$ is appropriately chosen, this scheme requires only *one* disk access per exact match query, and the search is about 33 times faster than infini-gram (Liu et al., 2024), a well-known corpus search engine based on suffix arrays.

**Example.** An example is shown in Figure 3. To find the position of "pattern match", we first search $Y$ and determine that the position lies between indices 3 and 6. We then search $X$ and find that the exact position is index 4.

Moreover, we reduce the index size via **run-length compression**; in our experience, this compression reduced the overall index size from $56.0\,\text{TB}$ to $21.6\,\text{TB}$ for the FineWeb-Edu corpus. We discuss the details in §H.2.

### 3.5. Soft Similarity Measure

We propose a new definition of pattern similarity $\mathsf{sim}(\bar{q}, \bar{p})$ to improve search quality.

**Substitutions: Smooth Minimum.** First, the basic similarity (without deletion or insertion) between a query

---

[3] Although BigTable employs a two-staged indexing scheme similar to ours, it is not based on suffix arrays and is similar to ours only at a high level.

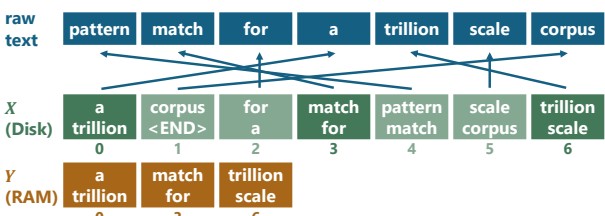

*Figure 3.* A sketch of our fast suffix array for the text "pattern match for a trillion scale corpus", with $L = 2$ and $B = 3$.

$\bar{q} = q_1 q_2 \cdots q_m$ and a pattern $\bar{p} = p_1 p_2 \cdots p_m$ is defined as follows, letting $c_i (\leq 1)$ be the cosine similarity between the word vectors of $q_i$ and $p_i$:

$$1 - \log_\beta \left( \sum_i (\beta^{1-c_i} - 1) + 1 \right).$$

This is a *smooth* minimum of $c_1, c_2, \ldots, c_m$, with a parameter $\beta$, which is set to $\beta = 10^4$ in our experiments. This improves over the plain minimum $\min(c_1, \ldots, c_m)$ (i.e., the limit $\beta \to \infty$ of the smooth minimum) used by SoftMatcha, which problematically ignores the similarities among all words except the one with the lowest score.[4] See Appendix D for further discussion.

**Insertions and Deletions: Norm-Based Penalties.** We extend the similarity above to allow insertions and deletions by scaling the similarity by some factor less than 1 for each insertion or deletion. To keep the penalty for low-information words such as "the" and "of" small, we use the word vector norm after Zipfian whitening (Yokoi et al., 2024), which correlates strongly with information entropy (Oyama et al., 2023). Letting $v$ be the squared norm of the inserted or deleted token, the scaling factor is $\exp(-v/\gamma)$. Here, $m$ is the length of the original search query given to our algorithm, rather than the length $i$ of the prefixes given to sim in Algorithm 1. Thus, $m$ does not change throughout the algorithm. In our experiments, we set $\gamma = m\gamma'$ for $\gamma'$ such that the scaling factor is equal to $1/e$ when $m = 5$ and the inserted word has the 50th-lowest norm (e.g., $\gamma' \approx 21.70$ for FineWeb-Edu).[5]

### 3.6. Additional Design Choice: Adaptive Similarity Thresholding

Practically, it is often hard to find a similarity threshold $\alpha$ that is neither too tight nor too loose. To mitigate this, Soft-

---

[4] For example, the similarity between "gold medal" and "bronze medals" is approximately 0.62, if the cosine similarity of "gold" and "bronze" is 0.63 and that of "medal" and "medals" is 0.85.

[5] Precisely speaking, we take the maximum similarity over the multiple ways to perform insertions and deletions.

Matcha 2 adaptively sets the similarity threshold as follows. First, we set the number $K$ of matching patterns that we want. SoftMatcha 2 first enumerates the soft matches (i.e., solves the above problem) with some pre-determined initial $\alpha$. If fewer than $K$ soft matches are found, SoftMatcha 2 relaxes the threshold $\alpha$ (i.e., decreases its value) by some amount and enumerates the soft matches again. SoftMatcha 2 repeats this until at least $K$ soft matches are obtained or the threshold $\alpha$ becomes too low.

## 4. Theoretical Analysis

Why is our algorithm (§3) so efficient, as observed in §5? How robust is it across several parameters? To answer these questions, we theoretically analyze our algorithm, drawing on general observations about natural languages. Concretely, we (1) prove that the expected number of total exact lookups conducted by Algorithm 1 is sublinear with respect to the corpus size $|\mathcal{C}|$ (Theorem 1) and (2) estimate it more in detail (Theorem 2), under several assumptions on the corpus and natural language statistics.

To this end, we analyze a single invocation of Algorithm 1. The details, including the full proofs, can be found in Appendix N. Let $\mathcal{C}$ be the corpus and $\bar{q} = q_1 q_2 \cdots q_m$ the query, as specified in §2.2. For simplicity, we fix the similarity threshold $\alpha$ and the insertion/deletion parameter $\gamma$. Let $R_i$ and $R'_i$, computed in the $i$-th iteration of Algorithm 1, be the sets $\{w \in R'_i \mid w \text{ occurs in } \mathcal{C}\}$ and $\{w \cdot w' \mid w \in R_{i-1}, w' \in \mathcal{V}^*, \text{sim}(q_1 q_2 \cdots q_i, w \cdot w') \geq \alpha\}$ respectively.

Now we analyze the total number of exact lookups in Algorithm 1, denoted $Total$. Crucially, it satisfies $Total = \sum_{i=1}^m |R'_i|$. We also let $\bar{\mathcal{C}}_i$ be $\{w \in \mathcal{V}^i \mid w \text{ occurs in } \mathcal{C}\}$, the set of vocabulary $i$-grams in the corpus.

For analysis, we assume the following bound on $|R'_i|$:

**Assumption 1** (Bound $|R'_i|$ by $|R_{i-1}|$)**.** There exists a constant $A$, independent of $\mathcal{C}$, such that, for any $i > 0$, $\mathbf{E}[|R'_i|] \leq A \mathbf{E}[|R_{i-1}|]$.

Intuitively, this is justified by assuming that the average number of $w' \in \mathcal{V}^*$ we explore for each $w \in R_{i-1}$ in calculating $R'_i$ is bounded by $A$. Empirically, the constant $A$ depends considerably on the similarity threshold $\alpha$.

First, if we assume Zipf's law of the $n$-gram distribution, we can analyze the effect of the corpus size $|\mathcal{C}|$.

**Theorem 1** (Sublinearity over the corpus size $|\mathcal{C}|$)**.** *Fix $m$. Assume that the frequency distribution of $n$-grams satisfies Zipf's law with exponent $\delta > 1$ for each $n$. Assume also that $\mathbf{E}[|R_i|] = O(\mathbf{E}[|\bar{\mathcal{C}}_i|])$ with respect to $|\mathcal{C}|$. Then the expected number of exact lookups in Algorithm 1 is approximately $O(|\mathcal{C}|^{1/\delta})$ with respect to the corpus size $|\mathcal{C}|$.*

*Table 3.* Corpora used in our experiments. For German, Italian, and French, we used subsampled datasets.

| Lang. | Corpus | # Tokens | Tokenizer | Embedding |
|---|---|---|---|---|
| English (EN) | FineWeb-Edu (Lozhkov et al., 2024) | **1,375,278,465,557** 11,186,491,048 | Moses (Koehn et al., 2007) LLaMA (Touvron et al., 2023) | GloVe (Pennington et al., 2014) LLaMA (Touvron et al., 2023) |
| Japanese (JA) | C4 (Raffel et al., 2020) | 168,756,125,544 | MeCab (Kudo et al., 2004)[6] | fastText (Grave et al., 2018) |
| Chinese (ZH) | C4 (Raffel et al., 2020) | 38,313,934,910 | ICU (The ICU Project, 2024) | fastText (Grave et al., 2018) |
| German (DE) | C4 (Raffel et al., 2020) | 1,099,885,019 | ICU (The ICU Project, 2024) | fastText (Grave et al., 2018) |
| Italian (IT) | C4 (Raffel et al., 2020) | 1,070,947,829 | ICU (The ICU Project, 2024) | fastText (Grave et al., 2018) |
| French (FR) | C4 (Raffel et al., 2020) | 978,329,060 | ICU (The ICU Project, 2024) | fastText (Grave et al., 2018) |
| Russian (RU) | C4 (Raffel et al., 2020) | 1,006,500,018 | ICU (The ICU Project, 2024) | fastText (Grave et al., 2018) |

Empirically, $\delta$ is typically 1.5 or so for many languages (Bellina & Servedio, 2025; Silva & Cunha, 2020). The assumption $\mathbf{E}[|R_i|] = O\big(\mathbf{E}[|\bar{\mathcal{C}}_i|]\big)$ is justified because most patterns in $R_i$ are in $\bar{\mathcal{C}}_i$. Indeed, it always holds (i.e., we do not need this as an assumption) if there are no insertions and deletions (by $\gamma \downarrow 0$), since then $R_i \subseteq \bar{\mathcal{C}}_i$ holds.

*Proof Sketch.* From Zipf's law, we can show that $\mathbf{E}[|\bar{\mathcal{C}}_i|]$, the expected number of distinct $i$-grams observed in $\mathcal{C}$, is approximately $O\big(|\mathcal{C}|^{1/\delta}\big)$ (Heaps' law). Then, since $|R_i|$ is bounded using $|\bar{\mathcal{C}}_i|$, $|R_i'|$ is bounded using $|R_{i-1}|$ by Assumption 1, and *Total* equals the sum of $|R_i'|$, the claim follows. See §N.1 for the full proof. □

Second, we estimate *Total* more in detail. We assume an exponential bound on $|R_i|$:

**Assumption 2** (Exponential bound on $|R_i|$)**.** There exist constants $a > 0$ and $r > 0$, independent of $i$, such that $\mathbf{E}[|R_i|] \le ar^i$ for any $i$.

Typically, the rate $r$ is expected to be near 1, but it depends on the choice of the similarity threshold $\alpha$ and the insertion/deletion parameter $\gamma$ (see Appendix M).

**Remark 1.** The model of Assumption 2 is intuitively justified by the following discussion from a linguistic perspective. Let $S_i$ be $\{w \in \mathcal{V}^* \mid \mathrm{sim}(q_1 q_2 \cdots q_i, w) \ge \alpha\}$, the set of vocabulary patterns that are similar to the $i$-gram prefix of the query (according to the threshold $\alpha$). We can suppose that its expected size $\mathbf{E}[|S_i|]$ is bounded from above by $a'\sigma^i$ for some constants $a'$ and $\sigma$, because $S_i$ is approximately the direct product of the sets of tokens similar to each token in the query. Also, let $\rho_i$ be $|\bar{\mathcal{C}}_i|/|\mathcal{V}|^i$, the corpus occurrence ratio over all vocabulary $i$-grams, which is typically very small for large $i$ since $|\bar{\mathcal{C}}_i| \le |\mathcal{C}|$. We can further suppose that $|R_i|$ is approximately the order of $\rho_i|S_i|$ because, since $R_i = \{w \in S_i \mid w \text{ occurs in } \mathcal{C}\}$, the occurrence of the query-similar patterns $S_i$ in the corpus $\mathcal{C}$ is expected to be close to uniform sampling according to $\rho_i$. However, we empirically observe that, as $i$ grows, the actual $|R_i|$ becomes much larger than $\rho_i|S_i|$ (see Appendix M for details); this is probably because $\bar{\mathcal{C}}_i$ is governed by linguistic constraints stronger than $R_i$, such as grammar and collocations.

By Assumption 2, we can make a more detailed estimate:

**Theorem 2** (Estimate under Assumption 2). *The expected number of exact lookups in Algorithm 1 is bounded from above by $Aa(1 - r^m)/(1 - r)$ when $r \ne 1$.*

If $r < 1$, the bound remains $O(1)$ with respect to $m$. Compared to $S_i$, the pruned candidate set $R_i$ dramatically mitigates exponential growth.

*Proof Sketch.* By straightforward calculation. See §N.2 for the full proof. □

## 5. Empirical Evaluation

### 5.1. Experimental Setup

**Corpus, Tokenizers, Word Vectors.** We used corpora of 7 different languages listed in Table 3. For English, we used FineWeb-Edu (Lozhkov et al., 2024), one of the largest publicly available corpora, comprising over one trillion tokens. Details are provided in Appendix C.

**Search Query.** We used (1) semi-automatically generated datasets using Gemini 3.0 Pro, and (2) real-world search queries (Craswell et al., 2020). The main text only includes the results of (1). See §H.4 for the results of (2). For the Gemini-generated dataset, we used 400 (English) and 100 queries (other languages) of token lengths of 10 or less. The queries covered various genres: see Appendix G for details, including the query generation method.

**Environment.** We conducted all experiments on an AWS `i4i.32xlarge` instance (128 vCPU, 1024GB RAM, 30TB NVMe SSD).

**Baseline.** We selected state-of-the-art implementations for exact lookup and soft search as of this writing as baselines. For exact lookup, we used infini-gram (Liu et al., 2024) and infini-gram mini (Xu et al., 2025); for soft search, we used SoftMatcha (Deguchi et al., 2025).

**Hyperparameters.** In the soft-search experiments, we retrieved the top $K = 20$ patterns, ranked by similarity to the query.[7] We only returned patterns with similarity $\ge 0.45$,

---

[6] https://taku910.github.io/mecab/

[7] For SoftMatcha (baseline), we also retrieved the top $K = 20$

because matches of very low similarity are usually irrelevant in practice (we denote the threshold $0.45$ as $\alpha$). Only 38 of the 400 English queries ($\approx 10\%$) returned fewer than $K$ results. Note that in our experiment, we used many hyperparameters: $K, \alpha, \beta,$ and $\gamma$ (some are written in §3.5), but the rationales for the values we set are shown in Appendix D.

**Web Application.** In this study, we also developed a web tool that actually searches corpora for up to hundred-billion-scale corpora. See Appendix B for details.

## 5.2. Qualitative and Linguistic Evaluation

Table 4 shows the results when we searched FineWeb-Edu for (1) "olympics gold medalist" and (2) "importance of the machine learning". In the first example, SoftMatcha 2 correctly found similar word sequences, such as "olympics silver medalist," and captured subtle differences in both meaning and spelling. In the second example, SoftMatcha 2 also retrieved sequences whose token count differs from the query, such as "importance of machine learning". Such variable-length matches were not supported by the original SoftMatcha. The results for other languages (e.g., Chinese) appear in Appendix E. In addition, we measured the output quality compared to baseline methods using LLM-as-a-judge. Compared to SoftMatcha, our method performed better in 61.3% of the cases. Details are shown in §I.1.

## 5.3. Disk Access in Exact Lookup

First, our method performed well in exact lookup, due to the low number of disk accesses. As shown in Figure 4, the p95 (95th percentile) latency for FineWeb-Edu (1.4T tokens) was only 0.34 ms. Compared to infini-gram (11.05 ms), our method was 33 times faster. For infini-gram mini, we were unable to conduct experiments with the full corpus (1.4T tokens) because index construction timed out, [8] but at 273B tokens, our method was 475 times faster (0.32 ms vs. 151.84 ms). This result demonstrates that our fast suffix array is useful on its own, in addition to our soft search.

## 5.4. Latency for Soft Search

Figure 5 shows the p95 latency in corpora of various languages and sizes. For FineWeb-Edu (English), our method achieved a median latency of 89.59 ms and a p95 latency of 278.17 ms even for 1.4T tokens, which is sufficiently practical for use as a corpus search tool. In addition, our method also achieves p95 latencies of less than 400 ms for Japanese and Chinese. Compared with SoftMatcha, latency

---

sequences. Note that SoftMatcha's implementation outputs all patterns with similarity above a specified threshold $\alpha$. To ensure a fair comparison, we first computed the similarity of the $K$-th ranked result as the threshold $\alpha$ **prior to the timed search**, and then used $\alpha$ to perform the search.

[8] At 273B tokens, the duration was 22.2 hours.

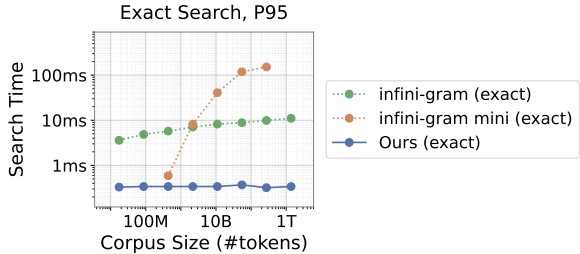

*Figure 4.* The p95 (95th-percentile) latency of exact search for FineWeb-Edu dataset (1.4T tokens). infini-gram mini timed out during index construction for larger corpora, and an error occurred for smaller corpora.

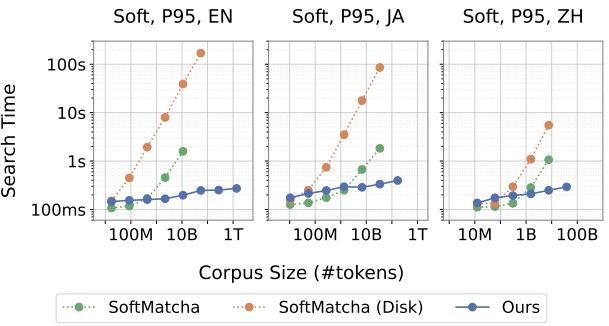

*Figure 5.* The p95 (95th-percentile) latency of soft search for EN (FineWeb-Edu, 1.4T tokens), JA (C4 Japanese, 169B tokens), and ZH (C4 Chinese, 38.3B tokens). SoftMatcha hit a memory limit, timed out during index construction, or encountered errors for larger corpora.

was nearly the same up to 500M tokens, but our method scaled better to larger corpora. Beyond 1B tokens, the latency of SoftMatcha increases almost linearly[9]. In contrast, our method showed almost no increase. In addition, our method also works well on real-world queries, not only on Gemini-generated queries (e.g., p95 latency for ORCAS was 433.50 ms). The detailed results, including (1) median latencies, (2) latencies for various $\alpha$ and $K$, and (3) latencies in some real-world queries, appear in Appendix H.

## 5.5. Disk Usage and Index Construction

For the full corpus of FineWeb-Edu (1.4T tokens, 6.7TB), the index of our SoftMatcha 2 is 21.6TB, *including* raw text. This is only 2.2 times greater than infini-gram's index size (9.9TB, *excluding* raw text), even though infini-gram supports only exact search. For index construction, our method took 53.8 hours, which was faster than infini-gram (61.2 h), infini-gram mini (timeout), and SoftMatcha (timeout). Although index construction is time-consuming, it is required only once, and we believe that it would not be a major problem in practice. Details are shown in §H.2.

---

[9] SoftMatcha was unable to handle corpora exceeding 50B tokens due to memory limits and timeout. See Appendix G for details.

*Table 4.* Example search results for FineWeb-Edu (English, 1.4T tokens) using the proposed method. Numbers in parentheses indicate hit counts; *1 denotes a SoftMatcha hit; *2 denotes an infini-gram (including infini-gram mini) hit. See Appendix F for the details on *2.

| # | Query: olympics gold medalist | | | | Query: importance of the machine learning | | | |
|---|---|---|---|---|---|---|---|---|
| | Pattern | Sim. | *1 | *2 | Pattern | Sim. | *1 | *2 |
| 1 | olympics gold medalist (838) | 100.0% | ✓ | ✓ | importance of the machine learning (103) | 100.0% | ✓ | ✓ |
| 2 | olympics gold **medallist** (456) | 89.1% | ✓ | ✗ | importance of ~~the~~ machine learning (12,662) | 85.4% | ✗ | ✗ |
| 3 | **olympic** gold medalist (95,816) | 86.4% | ✓ | ✗ | importance **,** of the machine learning (14) | 84.2% | ✗ | ✗ |
| 4 | olympics **silver** medalist (500) | 84.8% | ✓ | ✗ | **significance** of the machine learning (14) | 76.6% | ✓ | ✗ |
| 5 | **olympic** gold **medallist** (22,506) | 82.0% | ✓ | ✗ | importance ~~of~~ the machine learning (1) | 75.1% | ✗ | ✗ |
| ⋮ | | | | | | | | |
| 20 | **paralympics** gold medalist (196) | 65.6% | ✓ | ✗ | importance of **all** machine learning (12) | 56.2% | ✓ | ✗ |

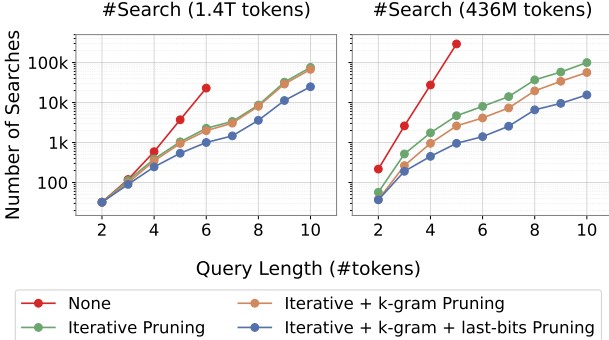

*Figure 6.* The geometric mean number of exact string matching lookups with and without enabling the pruning techniques over the FineWeb-Edu dataset (1.4T tokens and subsampled 436M tokens). No data is displayed if a timeout (10 sec.) occurred.

### 5.6. Ablation and Sensitivity Analysis

We evaluated the effectiveness of the three pruning techniques explained in §3.3, i.e., (1) iterative-pruning, (2) $k$-gram pruning, and (3) last-bits pruning. Figure 6 shows the geometric mean number of exact string matching lookups for FineWeb-Edu (1.4T-token dataset and 436M-token subsampled dataset). On average, it increases 2.30 times per token with all three pruning techniques, while 5.17 times with none of them, demonstrating the effectiveness of the proposed techniques. See §K.3 for further discussion.

We also analyze the sensitivity of search quality to the choice of hyperparameter value $\beta$. The details are shown in §I.4.

## 6. Practical Applications

### 6.1. Contamination Detection

Evaluating LLMs intrinsically is extremely difficult due to their complexity; therefore, extrinsic evaluation using a benchmark dataset of hard problems is essential for the research and development of LLMs. Therefore, contamination of benchmark (i.e., test set) data by information from the pre-training corpus (i.e., training set) is harmful for the research and development of LLMs.

As a practical case study, this section demonstrates the effectiveness of our method in benchmark contamination detection. Although infini-gram mini (Xu et al., 2025) has also been applied for the same purpose and identified contaminations in 24 popular benchmarks, it relies solely on exact matching. By applying soft search, we can detect two critical forms of leakage that are overlooked by exact matching: (1) **Semantic contamination**, in which a token sequence that is semantically similar to a benchmark task is found in a pre-training corpus; and (2) **Template leakage**, in which a token sequence that is obtained by substituting a different numerical expression into a benchmark task is found in a pre-training corpus (e.g., "Alice has 8 pens" and "Alice has 5 pens").

**Contamination Detection Method.** Inspired by the experiments in infini-gram mini (Xu et al., 2025), we extract all 10-token substrings (instead of 50 characters in infini-gram mini), compute

$$\eta = \frac{1}{|S|} \sum_{s \in S} \text{softmatch}(s),$$

where $\text{softmatch}(s)$ is 1 if a substring with similarity $\geq 0.6$ exists in the corpus, and 0 otherwise. If $\eta \geq 0.8$, then a benchmark task is marked *dirty*.

**Benchmarks and Corpus.** We analyze 7 benchmarks that are also used in infini-gram mini experiments (Xu et al., 2025). The benchmarks consist of problems taken from various categories: (1) knowledge and reasoning, (2) math, (3) coding, and (4) common-sense understanding. The full list of benchmarks is given in Appendix O. For large test sets, we randomly sampled 400 entries. For benchmarks with questions and answers, we use only the question part as patterns. As a pre-trained corpus, we used the full FineWeb-Edu dataset (1.4T tokens).

**Verifying True Contamination.** We manually inspected each problem flagged as "dirty" by the soft search but not by the exact match (i.e., maximum similarity $< 1.0$ but $\geq 0.6$), to determine whether the problem is truly contaminated or not. Specifically, we classified each problem as (1) semantic contamination, (2) template leakage, or (3) false positive.

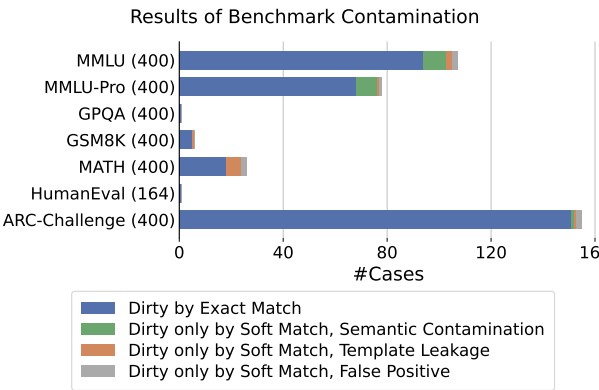

*Figure 7.* The number of problems flagged as dirty, including the breakdown of contamination. The numbers in brackets are the sample counts per dataset. For a dataset of more than 400 samples, we randomly subsampled to 400 samples.

**Quantitative Results.** Figure 7 shows the number of problems flagged as dirty by exact match and soft search, along with the breakdown of semantic contamination. In total, the standard exact match flagged 338 out of 2,564 samples (13.2%) as dirty. Our method detected an additional 36 samples (1.4%) that the exact match overlooked. Importantly, this manageable volume allowed for an exhaustive manual inspection, which revealed a high precision rate of 81% (29/36); 18 were classified as semantic contamination (e.g., comma insertions), and 11 were classified as template leakage (e.g., numerical substitution). The complete list, including the classification results, is shown in Appendix O.

Below, we show two interesting examples of contamination that we discovered. **Example 1**: In the MMLU benchmark, one question asks "The difference between dc and ac in electric circuits is that in dc, charges flow". In the corpus, we found that the text "The difference between DC and AC ⅲⅲ electric circuits is that in DC, charges flow in one direction." where "in" is simply deleted from the question text. This is an example of semantic contamination that can affect the fairness of the evaluation. **Example 2**: In the MATH benchmark, one question reads "The sum of two numbers is 25 and their difference is 11. What is the larger of the two numbers?". In the corpus, we found the text "The sum of two numbers is **139** while their difference is **19**. What is the larger of two numbers?" in the corpus; one is obtained by numerical substitution from the other. Furthermore, the solution to this problem immediately follows this text in the pre-trained corpus. This is an example of template leakage.

### 6.2. Other Tasks

We also conducted some evaluations for practical tasks: (1) information retrieval, and (2) paraphrase detection. The results are shown in Table 5, and our method outperformed or matched the baselines, including BM25, infini-gram, and

*Table 5.* The performance in information retrieval and paraphrase detection tasks.

| Method | Information Retrieval | | | | Para-phrase |
| --- | --- | --- | --- | --- | --- |
| | TREC-COVID | | SciFact | | |
| | P@20 | R@1000 | P@1 | R@5 | |
| BM25 | 33.8 | 14.9 | **42.7** | 58.1 | 82.8 |
| infini-gram | — | — | — | — | 65.5 |
| SoftMatcha | 35.4 | 17.7 | 42.3 | 60.5 | 74.5 |
| Ours | **36.0** | **18.0** | **42.7** | **60.7** | **84.3** |

SoftMatcha. The details are shown in §I.2 and §I.3.

## 7. Conclusion

We developed SoftMatcha 2, an ultra-fast and flexible algorithm for matching queries against trillion-scale natural language corpora under semantic variations, centered on dynamic pruning for semantic query relaxation, built on top of a newly designed, disk-aware suffix-array index. We theoretically analyze the efficiency of the proposed method, indicating how it can mitigate exponential growth in the search space by leveraging the statistical properties of natural language. Empirically, our method achieves soft matching at the 0.1-second level and substantially reduces latency compared with infini-gram, infini-gram mini, and SoftMatcha. As a practical outcome, we show that the proposed method enables information retrieval, paraphrase detection, and the detection of contamination between training corpora and evaluation data.

We also note several limitations of the proposed method. First, it operates at the word level and cannot capture semantic variations that span multiple words, such as the similarity between "U.S." and "United States". The theoretical analysis in §4 presupposes statistical regularities of natural language to effectively suppress the combinatorial explosion induced by semantic query relaxation. In addition, the proposed method requires memory and disk resources that scale with corpus size, which can make deployment and system setup demanding.

An important future direction is to strengthen the theoretical grounding of our analysis by collecting broader empirical evidence and relating it to known statistical regularities and universals of natural language. Another promising direction is to incorporate compositional representations, such as sums of word vectors, to better handle multi-word semantic variations; however, doing so would substantially increase the number of substitution candidates, making it challenging to preserve low-latency search. Supporting more general queries, such as wildcards, is also a future direction. Finally, we plan to extend the online demo to additional languages and corpora, enabling broader analysis of large-scale training data and facilitating community use.

## Acknowledgements

This work was supported by JSPS KAKENHI JP24KJ0133, JP23K24910, JP25H01113; JST CREST JPMJCR2012; JST FOREST JPMJFR2331; JST PRESTO JPMJPR22CA; JST BOOST JPMJBY24H8, JPMJBS2412; the Hakubi Project at Kyoto University; and the NINJAL BCCWJ2 Project commissioned by the Agency for Cultural Affairs, Japan. The second author is also hosted by MPI-SWS as a JSPS Overseas Research Fellow.

We are grateful to Hiroyuki Deguchi for his helpful remarks on the details of SoftMatcha (Deguchi et al., 2025). We would like to thank Zhenya Zhang, Étienne André, Benjamin Heinzerling, Paolo Arcaini, and Anna Lukina for commenting on the query examples in Chinese, French, German, Italian, and Russian, respectively. We would like to thank Yoshitaka Haribara for his support with the computing environment. We are grateful to Reina Akama for her advice on the public release of our tool. Finally, we thank the anonymous reviewers and meta-reviewers for their thoughtful and constructive feedback, which helped us improve the theory, experiments, and presentation of this paper.

## Impact Statement

This paper presents work whose goal is to advance the field of machine learning, natural language processing, and language models. There are many potential societal consequences of our work, none of which we feel must be specifically highlighted here.

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

# A. Additional Related Work

## A.1. Corpus Search for Information Retrieval, NLP, and Language Models

In corpus linguistics and quantitative linguistics, contextually grounded examples of words or phrases, known as KWIC (Key Word In Context; Luhn, 1960), are used with concordancers to analyze patterns of actual language usage (Lexical Computing, 2015; NINJAL, 2011; Berlin-Brandenburgische Akademie der Wissenschaften, 2026). In traditional natural language processing, corpus searching serves as a general subroutine for extracting world knowledge from text corpora (information extraction) or as a preprocessing step for analyzing product reviews (sentiment analysis). In addition, language models are fundamentally built from large-scale corpora, making the study of language models inseparable from corpus research (Dodge et al., 2021). For instance, crucial studies that identify harmful biases and inappropriate expressions in training corpora, or detect benchmark contamination, are grounded in corpus search (Guo et al., 2022; Ippolito et al., 2023; Deng et al., 2023; Xu et al., 2024). Moreover, with the discovery of scaling laws (Kaplan et al., 2020) and the exponential growth of training corpora (Gokaslan et al., 2019; Raffel et al., 2020; Penedo et al., 2024), corpora have reached sizes that are no longer manually verifiable, thereby increasing the necessity for efficient corpus searching methods in recent years (Liu et al., 2024; Xu et al., 2025; Deguchi et al., 2025).

It is also worth mentioning methods that share the objective of "retrieving documents from queries" but differ somewhat in their goals and primary purposes. In corpus linguistics and classical natural language processing, regular expressions have been used as versatile and powerful pattern matchers (Knight, 2013; Manning & Schütze, 1999). However, natural language frequently involves synonymous variants, phrase-level paraphrases, and various insertions and deletions of pronouns and adverbs, making it challenging to use regular expressions for large-scale corpus searches, such as contamination detection (Deng et al., 2023; Xu et al., 2024) that we discuss in §6. In the field of information retrieval, lexical search methods such as TF-IDF (Salton & Buckley, 1988) and BM25 (Robertson et al., 1994) have been widely used (Ceri et al., 2013). These methods use a form of set similarity between a query and a document to retrieve "related" documents, rather than performing strict text searches. Dense vector search (Karpukhin et al., 2020) is a more modern approach for finding "related" documents to a query. Technically, it uses text vectors (Reimers & Gurevych, 2019; Gao et al., 2021) and approximate nearest neighbor (ANN) search (Indyk & Motwani, 1998; Andoni & Indyk, 2008; Malkov & Yashunin, 2020), and has recently been particularly popular in Retrieval-Augmented Generation (RAG; Lewis et al., 2020).

## A.2. String Matching Algorithms

*String matching*, or matching a pattern string against a target string, has been a fundamental research topic in computer science with a long history; see, e.g., (Adjeroh et al., 2008) for the details. Traditionally, string matching has been designed to match strings at the letter level; however, most of the string-matching algorithms can be naturally applied to token-level search rather than letter-level search.

A popular approach for scalable string matching is to build an index of the target text in advance and use it at runtime. *Suffix arrays* (Manber & Myers, 1990) and *inverted indexing* (Zobel & Moffat, 2006) are representative indexing schemes. Using a suffix array, one can locate the occurrences of the query text in $O(\log|\mathcal{C}|)$ time, where $|\mathcal{C}|$ is the corpus size. In contrast, a search with an inverted index takes linear time w.r.t. the frequency of the rarest word in the target text—prohibitively costly for a massive corpus.

String matching is generalized for *approximate* matching (i.e., matching a pattern to a target text modulo substitutions, insertions, and deletions of letters), where the distance from the query text is measured by an edit distance (Wagner & Fischer, 1974). In addition to relaxation based on the *semantic* distance, our method supports matching with word insertions and deletions, inspired by this line of work.

# B. Web Interface

As noted in the abstract, we provide a web interface (demo tool) that works for corpora of up to hundreds of billions of tokens; see Figure 8. Figure 9 shows another screenshot of our demo. You can use the demo via the following link:

> https://softmatcha.github.io/v2/

# C. Tokenizers and Embeddings

**Details of Embeddings.** For English, we use `glove-wiki-gigaword-300` (Pennington et al., 2014) with a vocabulary size of 400,000. For other languages, we use `facebook/fasttext-{ja, zh, fr, de, it, ru}-vectors` (Grave et al., 2018) with a vocabulary size of 524,288. Although the original fastText models contain 2,000,000 words, we restrict the vocabulary to the top 524,288 entries to improve retrieval latency.

**Subword Tokenizer.** We also conducted an experiment with a model with subword tokenizer (see Appendix J for details). We selected `meta-llama/Llama-2-7b-hf` with a vocabulary size of 32,000 in this experiment.

**Compatibility.** The tokenizers and embeddings we used (e.g., Moses and Glove) are not completely compatible, but empirically they are compatible in most of the cases.

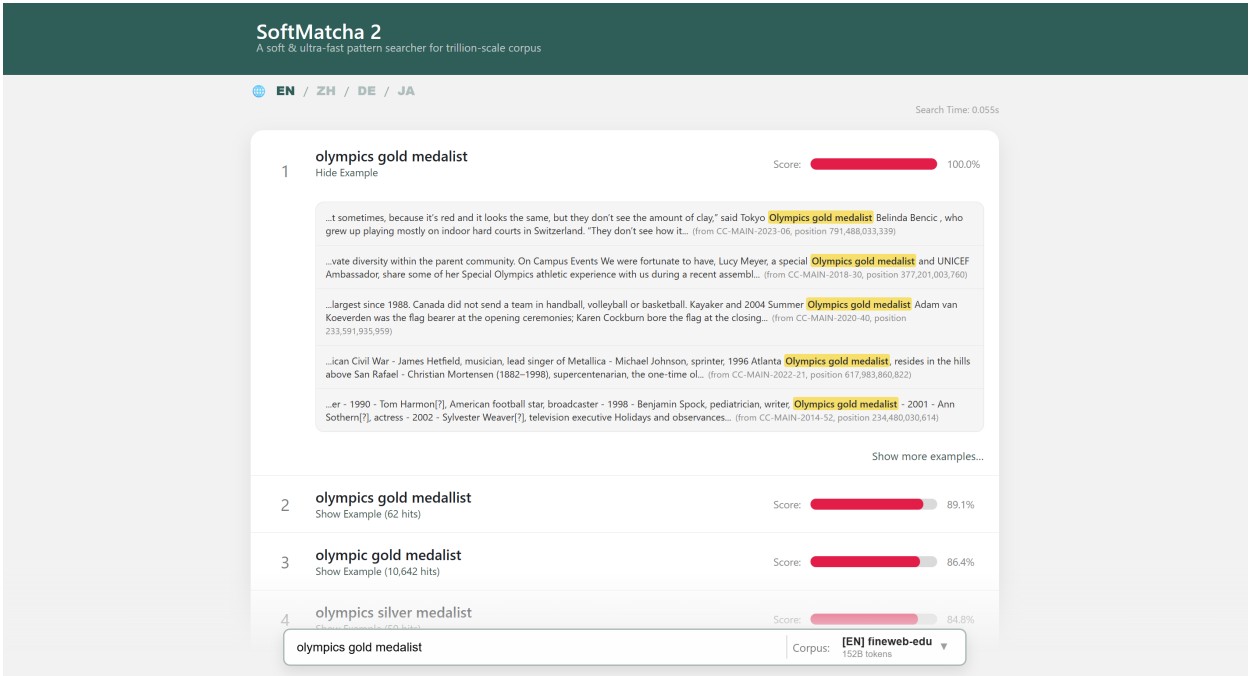

*Figure 8.* Screenshot of a search for "olympics gold medalist", showing example matches. In this example, the demo returns the top 20 patterns in 55 ms.

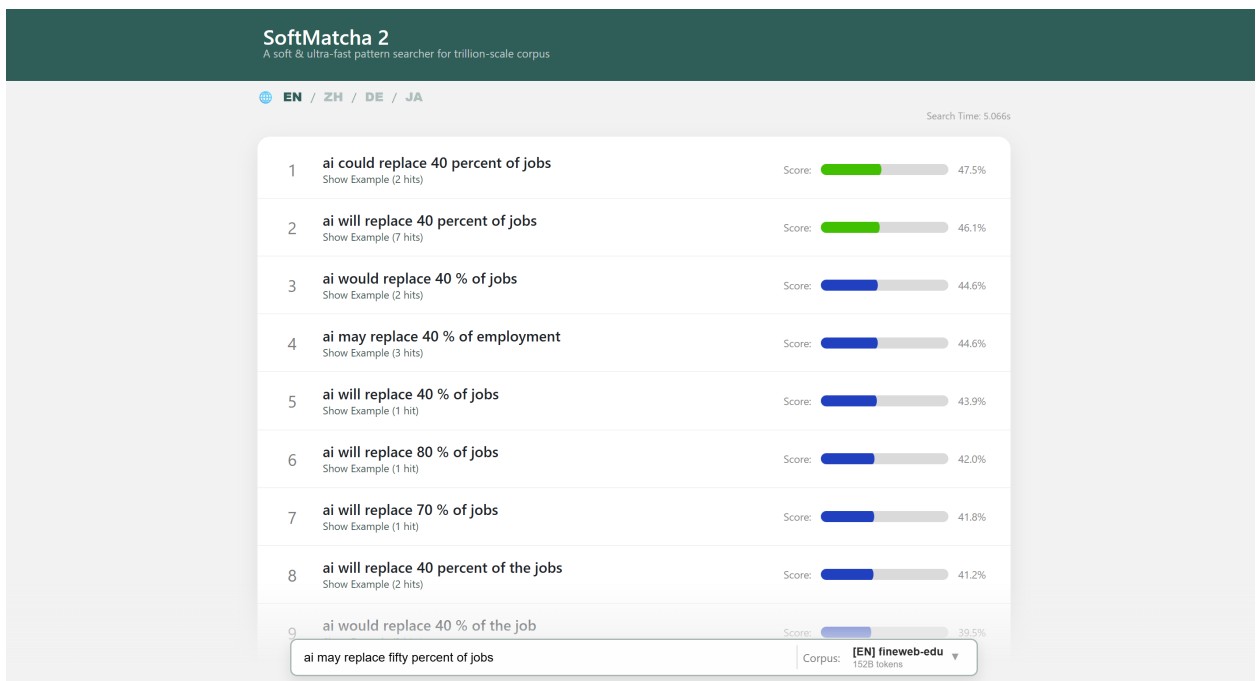

*Figure 9.* Screenshot of a search for "ai may replace fifty percent of jobs". Using extended search mode, the demo returns 40 patterns with similarity $\geq 0.27$.

## D. Rationales of the Hyperparameters

**Maximum Query Length $L$.** In the experiments, we set the maximum query length ($L = 12$), which is long enough for most use cases. Studies on real search workloads show that the vast majority of queries are short: in concordancer logs, more than 87% of queries are less than or equal to five words (Valli, 2012); in general online search, 99.9% of queries are less than or equal to twelve words (Bendersky & Croft, 2009). Macklovitch et al. (2008) report that 97.8% of search queries are 5 words or fewer. In our preliminary user study, more than 99% of queries were shorter than 11 words. Since the index size scales linearly with $L$, using a larger $L$ would substantially increase storage costs without providing significant benefit to most users.

**Number $K$ of Initial Search Results.** Our system presents the top $K$ search results. Thresholding by the number of results rather than the similarity follows the common practice in general retrieval/search systems such as Google Search, where users expect the top results to be returned quickly. We set $K = 20$ to balance retrieval efficiency with result coverage. We discuss the influence of the choice of $K$ on the efficiency in §H.1.

**Smooth Minimum Function.** We used the smooth minimum (softmin) function parameterized by $\beta$ to balance two conflicting goals: focusing on the most deviating words while still accounting for the others. On the one hand, plain *minimum* (as in SoftMatcha) ignores non-minimum similarities. For example, "gold medal" and "gold medals" have exactly the same score for the query "silver medal". On the other hand, plain *summation* over-penalizes multiple mismatches. For example, "Theorem 1" has a score closer to "Player 1" than to "Lemma 2". Our smooth minimum interpolates between these extremes via the parameter $\beta$. See §I.4 for the ablation on the value of $\beta$.

**Similarity Threshold $\alpha$.** We set $\alpha = 0.45$ so that approximately 90% of queries return the full $K = 20$ results. As stated in §5, only 38 of the 400 queries returned fewer than $K$ results. Lower thresholds produce unrelated matches. For example, the similarity of "natural language processing" and "natural language software" is already $0.446$.

## E. Multilingual Applicability

We also conducted some qualitative evaluations for Japanese, Chinese, French, German, Italian, and Russian. Tables 6 to 11 show the results along with translations to English, and all of them were able to capture semantic similarities and insertions/deletions.

### E.1. Japanese

Table 6 shows example search results for a Japanese corpus (C4 with 2.14B tokens) with a query "大阪までの2時間半" meaning "the two and a half hours to Osaka". We observe that all the search results have a similar meaning: they are about the time from or to a city in Japan (e.g., Kyoto). Notably, we observe that token removal also occurs in Japanese: in 大阪まで4時間半 (#1), the token "の" is removed, which makes "4時間半" a noun phrase.

### E.2. Chinese

Table 7 shows example search results for a Chinese corpus (C4 with 2.24B tokens). For Query 1 (从中国到美国的旅游, meaning "travel from China to the United States"), most search results follow the same template: 从 {place} 到 {place} (的) {travel-related noun}, where 从 means "from," 到 means "to," 的 marks a modifier, and 旅游/游客 mean "travel"/"tourists", with small variations such as replacing the origin/destination places (e.g., 从中国到美国的高铁 (#5), meaning "high-speed rail from China to the United States") or switching to a related relation such as "between" by using 和 ("and") (e.g., 从中国和美国的机票 (#4), meaning "air tickets between China and the United States").

For Queries 2-1 and 2-2 (我 中午 吃飯 的 and 我 中午 吃饭 的, both meaning "what I eat at noon"), most search results follow the same template: {*person*} {*time*} {*eat/sleep*} *(的)* (where 我/你/您 mean "I/you/polite you," 中午/晚上 mean "at noon/at night," 吃饭/吃飯 mean "eat (a meal)," and 睡觉/睡覺 mean "sleep"), with small variations such as changing the person (e.g., 你 中午 吃饭 的 (#2) of Table 7c, meaning "what you eat at noon"), changing the time and verb (e.g., 我 晚上 睡觉 的 (#4) of Table 7c, meaning "what I do at night: sleep"), or omitting the final particle 的 (e.g., 我 中午 吃飯 (#1) of Table 7b, meaning "I eat lunch at noon").

Queries 2-1 and 2-2 are identical in meaning but use different scripts: Query 2-1 uses traditional characters (e.g., 吃飯, 睡覺), whereas Query 2-2 uses simplified characters (e.g., 吃饭, 睡觉). This difference is reflected in the results: most matches for Query 2-1 are written in traditional characters, while most matches for Query 2-2 are written in simplified characters. At the same time, cross-script matching still occurs, for example, #2 of Table 7b matches a traditional-character query with a simplified-character pattern (我 中午 吃饭 的, meaning "what I eat at noon").

### E.3. French

Table 8 shows example search results for a French corpus (C4 with 978M tokens). For Query 1 (*je mange de la viande*, meaning "I eat meat"), most search results fol-

*Table 6.* Example search results for C4 (Japanese, subsampled 2.14B tokens) using the proposed method. Numbers in parentheses indicate hit counts.

*(a)* Query 1: 大阪までの 2 時間半 (meaning: "the two and a half hours to Osaka")

| # | Pattern | Translation | Similarity |
|---|---------|-------------|------------|
| 1 | 大阪 まで 4 時間 半 (1) | it takes four hours and a half to Osaka | 75.0% |
| 2 | 大阪 から 7 時間 半 (1) | it takes seven and a half hours from Osaka | 58.9% |
| 3 | 京都 まで が 2 時間 半 (1) | it takes two and a half hours to Kyoto | 57.4% |
| 4 | 神戸 から 、 2 時間 半 (1) | it takes two and a half hours from Kobe | 55.2% |
| 5 | 東京 まで 1 時間 半 (1) | it takes one and a half hours to Tokyo | 53.0% |
| ⋮ | | | |
| 20 | 名古屋 から 2 時間 半 (1) | it takes two and a half hours from Nagoya | 47.9% |

*Table 7.* Example search results for C4 (Chinese, subsampled 2.24B tokens) using the proposed method. Numbers in parentheses indicate hit counts.

*(a)* Query 1: 从中国到美国的旅游 (meaning: "travel from China to the United States")

| # | Pattern | Translation | Similarity |
|---|---------|-------------|------------|
| 1 | 从 中国 到 美国 旅游 (1) | travel from China to the United States | 78.3% |
| 2 | 从 欧洲 去 美国 的 游客 (2) | tourists traveling from Europe to the United States | 46.2% |
| 3 | 从 武汉 到 韩国 的 游客 (1) | tourists traveling from Wuhan to South Korea | 45.4% |
| 4 | 从 中国 和 美国 的 机票 (1) | air tickets between China and the United States | 44.2% |
| 5 | 从 中国 到 美国 的 高铁 (1) | high-speed rail from China to the United States | 44.1% |
| 6 | 从 中国 到 欧洲 的 运输 (1) | transportation from China to Europe | 43.8% |
| 7 | 从 日本 到 美国 的 机票 (2) | air tickets from Japan to the United States | 43.7% |

*(b)* Query 2-1: 我 中午 吃飯 的 (meaning: "what I eat at noon")

| # | Pattern | Translation | Similarity |
|---|---------|-------------|------------|
| 1 | 我 中午 吃飯 (1) | I eat lunch | 69.3% |
| 2 | 我 中午 吃饭 的 (9) | what I eat at noon | 63.2% |
| 3 | 你 中午 吃饭 的 (1) | what you eat at noon | 60.7% |
| 4 | 你 晚上 睡覺 的 (1) | what you do at night: sleep | 59.2% |
| 5 | 我 晚上 睡覺 ， (1) | I sleep at night, | 58.1% |
| ⋮ | | | |
| 20 | 您 晚上 吃饭 ， (2) | you eat dinner, (polite) | 49.9% |

*(c)* Query 2-2: 我 中午 吃饭 的 (meaning: "what I eat at noon")

| # | Pattern | Translation | Similarity |
|---|---------|-------------|------------|
| 1 | 我 中午 吃饭 的 (9) | what I eat at noon | 100.0% |
| 2 | 你 中午 吃饭 的 (1) | what you eat at noon | 76.8% |
| 3 | 我 中午 吃饭 (9) | I eat lunch at noon | 69.3% |
| 4 | 我 晚上 睡觉 的 (3) | what I do at night: sleep | 64.0% |
| 5 | 你 晚上 睡觉 的 (2) | what you do at night: sleep | 61.4% |
| ⋮ | | | |
| 20 | 自己 正午 吃饭 的 (1) | what oneself eats at noon | 53.9% |

*Table 8.* Example search results for C4 (French, subsampled 978M tokens) using the proposed method. Numbers in parentheses indicate hit counts.

*(a)* Query 1: *je mange de la viande* (meaning: "I eat meat")

| # | Pattern | Translation | Similarity |
|---|---------|-------------|------------|
| 1 | *je mange de la viande* (17) | I eat meat | 100.0% |
| 2 | *ne mange de la viande* (1) | (does) not eat meat | 69.7% |
| 3 | *je remange de la viande* (1) | I eat meat again | 68.9% |
| 4 | *je consomme de la viande* (1) | I consume meat | 63.2% |
| ⋮ | | | |
| 13 | *je mange viande* (1) | I eat meat (with a grammatical error) | 58.0% |
| ⋮ | | | |
| 20 | *on mange de la nourriture* (1) | we eat food | 53.8% |

*(b)* Query 2: *de la vérification formelle* (meaning: "of formal verification")

| # | Pattern | Translation | Similarity |
|---|---------|-------------|------------|
| 1 | *de la vérification formelle* (4) | of formal verification | 100.0% |
| 2 | *la vérification formelle* (10) | the formal verification | 79.9% |
| 3 | *sa vérification formelle* (1) | its formal verification | 64.5% |
| 4 | *de la vérification rigoureuse* (1) | of rigorous verification | 55.8% |
| 5 | *vérification formelle* (16) | formal verification | 50.6% |
| ⋮ | | | |
| 15 | *et la vérification formels* (1) | and the formal verification(s) | 45.1% |

low the same template: {*subject*} *(ne)* {*eat*} *(pas) de la* {*food*}, where *je/on* mean "I"/"we," *mange* means "eat," *ne ⋯ pas* expresses negation, and *viande/nourriture* mean "meat"/"food", with small variations such as adding negation (e.g., *ne mange de la viande* (#2), meaning "(someone) does not eat meat") or replacing the verb with a near-synonym (e.g., *je consomme de la viande* (#4), meaning "I consume meat"). We also observe a grammatical error in #13: *je mange viande*; the grammatical form is *je mange de la viande*.

For Query 2 (*de la vérification formelle*, meaning "of formal verification"), most search results follow the same template: *({determiner}) vérification {modifier}*, where *de la/la/sa* mean "of the"/"the"/"its," and *formelle* means "formal", with small variations such as dropping the determiner (e.g., *vérification formelle* (#5), meaning "formal verification") or replacing the modifier with a semantically close one (e.g., *de la vérification rigoureuse* (#4), meaning "of rigorous verification").

Token insertion and removal occur in French, such as *de la vérification formelle* ("of formal verification"; Query 2) vs. *vérification formelle* ("formal verification"; #5 in Table 8b).

### E.4. German

Table 9 shows example search results for a German corpus (C4 with 1.10B tokens). For Query 1 (*der Bus von Graz nach Wien*, meaning "the bus from Graz to Vienna"), the search results largely preserve the same template: *(article)* {*Bus/Taxi/Flug*} *von* {*city*} *nach* {*city*} (*Bus/Taxi/Flug* mean "bus/taxi/flight," *von* means "from,"

and *nach* means "to"), while substituting the vehicle type and the origin/destination cities (e.g., #1 is "the bus from Vienna to Munich," and #20 is "the flight from Graz to Innsbruck").

For Query 2 (*Ich liebe Bücher aber*, meaning "I love books, but"), most results keep the prefix *ich liebe Bücher* ("I love books") and vary the continuation by inserting or replacing a few short words: *also* ("so"; #1), *die* ("that/which"; #3), and *und auch* ("and also"; #4). In contrast, #6 changes the polarity by replacing *liebe* ("love") with *hasse* ("hate"), yielding *ich hasse Bücher* ("I hate books").

Token insertion and removal occur in German, such as *ich liebe Bücher aber* ("I love books, but"; Query 2) vs. *ich liebe Bücher , also* ("I love books, so"; #1 in Table 9b).

### E.5. Italian

Table 10 shows example search results for an Italian corpus (C4 with 1.07B tokens). For Query 1 (*di roma e di napoli*, meaning "of Rome and of Naples"), all search results follow the same template: *(di)* {*city*} *(,) e (di)* {*city*}, where *di* means "of" and *e* means "and", with small variations such as optional punctuation insertion (e.g., *di roma , e di napoli* (#2), meaning "of Rome, and of Naples") or dropping the first *di* (e.g., *roma e di napoli* (#3), meaning "Rome and of Naples").

For Query 2 (*generalmente di proprietà di*, meaning "generally owned by"), most search results follow the same template: {*adverb of frequency*} *di proprietà (di)*, where *generalmente* means "generally," *spesso* means "often," *normalmente* means "normally," *sempre* means "always" and

*Table 9.* Example search results for C4 (German, subsampled 1.10B tokens) using the proposed method. Numbers in parentheses indicate hit counts.

*(a)* Query 1: *der Bus von Graz nach Wien* (meaning: "the bus from Graz to Vienna")

| # | Pattern | Translation | Similarity |
|---|---|---|---|
| 1 | *der Bus von Wien nach München* (1) | the bus from Vienna to Munich | 54.5% |
| 2 | *dem Taxi von Graz nach Wien* (1) | the taxi from Graz to Vienna | 52.4% |
| 3 | *den Bus von Wien nach Berlin* (1) | the bus from Vienna to Berlin | 51.9% |
| 4 | *dem Bus von Augsburg nach Wien* (1) | the bus from Augsburg to Vienna | 51.1% |
| 5 | *der Bus von Innsbruck nach Karlsruhe* (1) | the bus from Innsbruck to Karlsruhe | 48.7% |
| ⋮ | | | |
| 20 | *dem Flug von Graz nach Innsbruck* (1) | the flight from Graz to Innsbruck | 41.9% |

*(b)* Query 2: *Ich liebe Bücher aber* (meaning: "I love books, but")

| # | Pattern | Translation | Similarity |
|---|---|---|---|
| 1 | *ich liebe Bücher , also* (1) | I love books, so | 63.7% |
| 2 | *ich liebe Bücher ,* (6) | I love books, | 62.4% |
| 3 | *ich liebe Bücher die* (1) | I love books that | 56.5% |
| 4 | *ich liebe Bücher und auch* (1) | I love books and also | 54.5% |
| ⋮ | | | |
| 6 | *ich hasse Bücher ,* (1) | I hate books, | 53.8% |
| ⋮ | | | |
| 20 | *ich liebe Katzen auch* (2) | I love cats too | 48.7% |

*Table 10.* Example search results for C4 (Italian, subsampled 1.07B tokens) using the proposed method. Numbers in parentheses indicate hit counts.

*(a)* Query 1: *di roma e di napoli* (meaning: "of Rome and of Naples")

| # | Pattern | Translation | Similarity |
|---|---|---|---|
| 1 | *di roma e di napoli* (13) | of Rome and of Naples | 100.0% |
| 2 | *di roma , e di napoli* (1) | of Rome, and of Naples | 87.5% |
| 3 | *roma e di napoli* (13) | Rome and of Naples | 79.5% |
| 4 | *di roma e napoli* (70) | of Rome and Naples | 79.5% |
| 5 | *di roma , di napoli* (4) | of Rome, of Naples | 79.3% |
| ⋮ | | | |
| 20 | *di napoli e di roma* (7) | of Naples and of Rome | 68.6% |

*(b)* Query 2: *generalmente di proprietà di* (meaning: "generally owned by")

| # | Pattern | Translation | Similarity |
|---|---|---|---|
| 1 | *generalmente di proprietà di* (2) | generally owned by | 100.0% |
| 2 | *generalmente di proprietà* (284) | generally owned | 75.0% |
| 3 | *spesso di proprietà di* (6) | often owned by | 71.3% |
| 4 | *normalmente di proprietà* (1) | normally owned | 60.4% |
| 5 | *generalmente proprietà* (4) | usual characteristic | 56.3% |
| ⋮ | | | |
| 20 | *sempre di proprietà del* (4) | always owned by the | 46.3% |

*Table 11.* Example search results for C4 (Russian, subsampled 1.01B tokens) using the proposed method. Numbers in parentheses indicate hit counts.

*(a)* Query 1: золотую медаль на зимних олимпийских играх (meaning: "a gold medal at the Winter Olympic Games")

| # | Pattern | Translation | Similarity |
|---|---------|-------------|------------|
| 1 | золотую медаль на зимних олимпийских играх (3) | a gold medal at the Winter Olympic Games | 100.0% |
| 2 | бронзовую медаль на зимних олимпийских играх (1) | a bronze medal at the Winter Olympic Games | 68.7% |
| 3 | золотые медали на зимних олимпийских играх (1) | gold medals at the Winter Olympic Games | 45.2% |

*(b)* Query 2: Я очень люблю кофе (meaning: "I really love coffee")

| # | Pattern | Translation | Similarity |
|---|---------|-------------|------------|
| 1 | я очень люблю кофе (3) | I really love coffee | 100.0% |
| 2 | я очень люблю чай (1) | I really love tea | 77.8% |
| 3 | я очень люблю какао (1) | I really love cocoa | 74.2% |
| 4 | я очень люблю шоколад (1) | I really love chocolate | 65.2% |
| 5 | я очень люблю лимонад (1) | I really love lemonade | 56.3% |
| ⋮ | | | |
| 20 | но очень люблю пиво (1) | but I really love beer | 45.8% |

the final optional *di* means "by", with small variations such as replacing the adverb (e.g., *spesso di proprietà di* (#3), meaning "often owned by") or omitting the final *di* (e.g., *generalmente di proprietà* (#2), meaning "generally owned"). However, for #5, the meaning is a little bit different. The word "*proprietà*" has two meanings ("ownership" and "characteristic") and is used in this context in the latter sense.

Token insertion and removal also occur in Italian, such as *di roma , e di napoli* (#2 of Table 10a) and *generalmente di proprietà* (#2 of Table 10b).

### E.6. Russian

Table 11 shows example search results for a Russian corpus (C4 with 1.01B tokens). For Query 1 (золотую медаль на зимних олимпийских играх, meaning "a gold medal at the Winter Olympic Games"), all search results follow the same template: {*medal type*} медаль на зимних олимпийских играх (where золотую/бронзовую mean "gold"/"bronze," and медаль means "medal"), with small variations such as changing the medal type (e.g., бронзовую медаль на зимних олимпийских играх, #2, meaning "a bronze medal at the Winter Olympic Games") or changing number (e.g., золотые медали на зимних олимпийских играх, #3, meaning "gold medals at the Winter Olympic Games").

For Query 2 (Я очень люблю кофе, meaning "I really love coffee"), most search results follow the same template: я очень люблю {drink/food} (where я means "I," очень means "really/very," люблю means "love," and кофе/чай/какао mean "coffee"/"tea"/"cocoa"), with small variations such as substituting the object (e.g., я очень

люблю чай, #2, meaning "I really love tea") or replacing it with another beverage/food item (e.g., я очень люблю лимонад, #5, meaning "I really love lemonade"). In contrast, #20 adds a discourse marker но ("but") and drops the explicit subject, yielding но очень люблю пиво (meaning "but (I) really love beer").

## F. Comparison with Exact Match

Exact-match algorithms such as infini-gram and infini-gram mini often fail to find any matches, especially for long query sequences. In contrast, our soft searcher can still return some matches in these situations. Figure 10 shows the zero-hit ratio (i.e., the percentage of test data where no match was found) for exact and soft search with a similarity threshold of 0.45 on the FineWeb-Edu dataset. At 1.4T tokens, the zero-hit ratio for soft search was only 2.3% (9/400), compared with 5.8% (23/400) for exact search.[10]

## G. Additional Details of Experiments

**Executions.** To reproduce the same environment as the demo interface, for each experiment, we first clear the OS page cache and then sequentially run 400 search queries (or 100 queries for languages other than English). After finishing each query, we **did not** clear the OS page cache. This means that the operating system may retain cached data from previous queries.

---

[10] 5.8% refers to exact searches using our method. Since infini-gram is case-sensitive and distinguishes word-initial tokens from word-internal tokens, infini-gram's zero-hit ratio was considerably higher (e.g., 12.3% for 1.4T tokens).

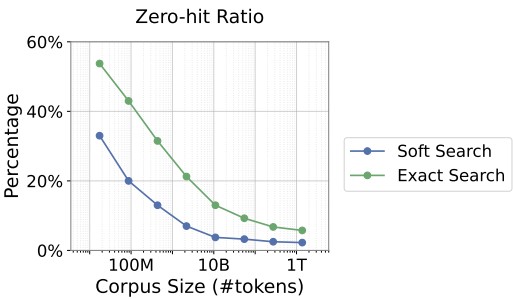

*Figure 10.* The zero-hit ratio (i.e., the percentage of test data where no match was found) for the FineWeb-Edu dataset on soft search and exact search, out of 400 English queries.

**Search Query.** We generated 400 English search queries using Gemini 3.0 Pro. We set the distribution of query lengths (in tokens) as shown in Table 13. For the English dataset, 70% of queries have 5 tokens[11] or less, while 20% of queries have 7 tokens or more. We chose this distribution because, while many practical example searches are likely to be short, searches for standard documents, such as stock-price articles, require handling longer queries. For Chinese, we used slightly shorter queries because the information density per token is higher than in English or Japanese. In terms of genres, we generated queries covering a wide range of topics, as shown in Table 12. The full list of queries is available at the following link:

> https://github.com/softmatcha/so
> ftmatcha2

The prompt used to generate queries with Gemini 3.0 Pro (translated from Japanese) is as follows:

> Please generate 400 English patterns that can be used for testing SoftMatcha [URL[12]]. Please follow the following distribution of query length: length 2: 5%, length 3: 20%, length 4: 25%, length 5: 20%, length 6: 10%, length 7: 7%, length 8: 5%, length 9: 5%, length 10: 3%. The topic should be as diverse as possible.

In addition, if the distribution is not as desired (e.g., too few basic phrases), we adjusted the distribution using follow-up prompts such as "please include more basic phrases".

**Raw Text.** To speed up index construction (including tokenization), we removed line breaks from the raw text so that each article occupies a single line, for all experiments, including SoftMatcha and infini-gram.

---

[11] Note that 5 tokens ≈ 25 characters.

[12] We used the previous SoftMatcha URL for generating queries (https://huggingface.co/spaces/softmatcha/wikitext-en).

*Table 12.* Percentage of queries in each topic category, as classified by Gemini 3.0 Pro. A query may belong to multiple categories.

| Topic | Count | % |
|---|---|---|
| Basic Phrases[13] | 79 | 19.8 |
| Proper Nouns | 231 | 57.8 |
| Science & Technology | 115 | 28.8 |
| Culture (Geography, History, etc.) | 213 | 53.3 |
| Politics | 158 | 39.5 |

*Table 13.* Distribution of token counts (in GloVe tokenizer) in queries used in the experiments.

| Query length | English | Japanese | Chinese |
|---|---|---|---|
| 1 tokens | 0% | 0% | 3% |
| 2 tokens | 5% | 5% | 22% |
| 3 tokens | 20% | 20% | 18% |
| 4 tokens | 25% | 25% | 17% |
| 5 tokens | 20% | 20% | 19% |
| 6 tokens | 10% | 10% | 8% |
| 7 tokens | 6% | 7% | 6% |
| 8 tokens | 6% | 5% | 6% |
| 9 tokens | 5% | 5% | 0% |
| 10 tokens | 3% | 3% | 1% |

**Notes on SoftMatcha.** SoftMatcha was unable to handle corpora exceeding 50B tokens due to memory limits. For example, a 1T-token corpus would theoretically require 20.0 TB of memory, making it unrealistic to handle such corpora. In addition, SoftMatcha also offers an execution mode that reads the index directly from SSD rather than expanding it in memory. However, if SoftMatcha is executed in this mode, the p95 latency was 169.89 seconds even with English 55B tokens, which was 688x slower than our proposed method. Note that for the SSD modes of SoftMatcha, we were unable to conduct experiments with larger corpora, because of a timeout. Specifically, the duration of index construction was projected to exceed 72 hours for a 273B-token corpus; at 55B tokens, the duration was 20.7 hours.

## H. Additional Quantitative Evaluations

### H.1. Additional Latency Evaluations

Figures 11 and 12 show the median latencies for exact match and soft search. Note that p95 latencies are already discussed in §5. In addition, we also measured latency for different numbers of outputs $K \in \{5, 10, 20, 40, 80\}$ and different similarity thresholds on the FineWeb-Edu dataset. Figure 13 shows the results. Even with $K = 80$ and a minimum similarity of 0.2, we achieved a median latency of 127.20 ms and a p95 latency of 5697.39 ms.

### H.2. Disk Usage

As described in §3.4, our implementation of the fast disk-aware suffix array applies the run-length compression to an index (array $X$ in Appendix L) to reduce disk space.

*Table 14.* The p95 latencies (ms) for cold cache and hot cache, across 400 Gemini queries.

| Method | Size | Cold Cache | Hot Cache |
|---|---|---|---|
| Ours | 8.6B | 252.85 | 187.30 |
|  | 273B | 280.22 | 243.39 |
| SoftMatcha (Disk) | 8.6B | 35,551.02 | 28,672.56 |
| Ours (Exact) |  | 0.87 | 0.35 |
| infini-gram (Exact) | 8.6B | 8.65 | 8.31 |
| infini-gram-mini (Exact) |  | 238.98 | 32.89 |

Without run-length compression, for FineWeb-Edu (1.4T tokens), it would require 40.0 TB of disk space to store the index and 56.0 TB in total. However, by applying run-length compression, the index size was reduced by 85.9% (7.1x; 40.0 TB → 5.6 TB), reducing total disk usage to 21.6 TB. Figure 14 shows the compression ratio of various languages and sizes.

Moreover, among the index of total size 21.6 TB, only 5.6 TB is accessed frequently to count soft-matched sequences, and the remaining 16 TB is accessed only to present example instances, which requires only about 10–20 random accesses per query. One possible system configuration is to store the hot part of the index (5.6 TB) on a high-speed NVMe SSD and store the remaining part (16 TB) on a RAID0 array of HDDs, which is realistic for a server-class machine. Assuming the random-access latency of the RAID0 array is about 10 ms, the additional latency would be only about 0.1–0.2 seconds per query.

### H.3. Cache Protocol

As described in Appendix G, our main experiments did not clear the OS page cache after each query (i.e., hot cache) to match the conditions of the demo tool. However, to verify whether our method also works with a cold cache, we also measured latencies when we clear the OS page cache, using the FineWeb-Edu subset of at most 273B tokens. The results are shown in Table 14. Except for infini-gram mini, which became drastically slower, the overall ranking did not change. In addition, our method for the soft and exact search was still >100x and >9x faster than the previous methods, respectively.

### H.4. Real Data Queries

Although we used Gemini-generated queries to test the runtime, we also conducted the experiment using the two types of real-world data: (1) ORCAS, which is the list of actual search queries related to Bing, and (2) random substrings extracted from FineWeb-Edu. For (1) ORCAS, we selected 1,000 random queries proportional to the number of documents (i.e., popular search words are picked more frequently than other words). In addition, we excluded queries whose

length is 11+ tokens or containing some unknown words. As a result, we used 827 queries. For (2) FineWeb-Edu, we used 1,000 queries with the same length distribution as Table 13.

The results are shown in Table 15; With a corpus of 8.6B subset of FineWeb-Edu, the p95 latency for both queries was less than 500 ms, which is within the same order of magnitude as Gemini queries.

## I. Additional Qualitative Experiments

### I.1. Quality Judging

First, we compared the quality of the output patterns among (a) SoftMatcha, (b) our method with plain minimum similarity, and (c) our method with smooth minimum similarity ($\beta = 10^4$), using LLM-as-a-judge. Since SoftMatcha uses the plain minimum and does not support insertion, the only difference between (a) and (b) is insertion. Specifically, we set $\alpha = 0.45$, $K = 20$, and used GPT-5.4 with the following prompt to compare two methods A and B:

> We are developing a search tool that retrieves and displays words that are exactly or partially similar to a given query string, along with their hit counts in a corpus. When we searched for the query "machine learning" using two different algorithms, we obtained the following results.
>
> > [Result A]
> > (Top-$K$ output for method A)
> > [Result B]
> > (Top-$K$ output for method B)
>
> Please evaluate the quality of [Result A] and [Result B] based on the semantic relevance and usefulness of the retrieved words to the query. Do not judge based on the number of matches. Rate the results by choosing one of the following options:
>
> > - 'A' (Result A is better)
> > - '=' (Both are of equal quality)
> > - 'B' (Result B is better)
>
> First, provide a brief justification for your choice, and then output your final answer as strictly 'A', '=', or 'B' on the last line.

To ensure fairness, we only used queries in which both methods output full $K$ patterns, because it is not ideal to be strongly biased with the number of outputs, which is a less important factor[14]. As a result, of the 400 Gemini-generated queries, we used 274 for the comparison between

---

[14] In fact, we can easily adjust the number of output patterns by slightly changing $\alpha$.

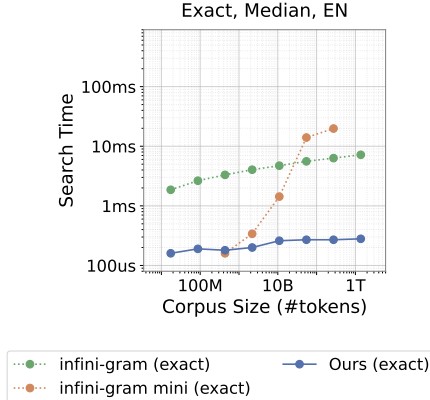

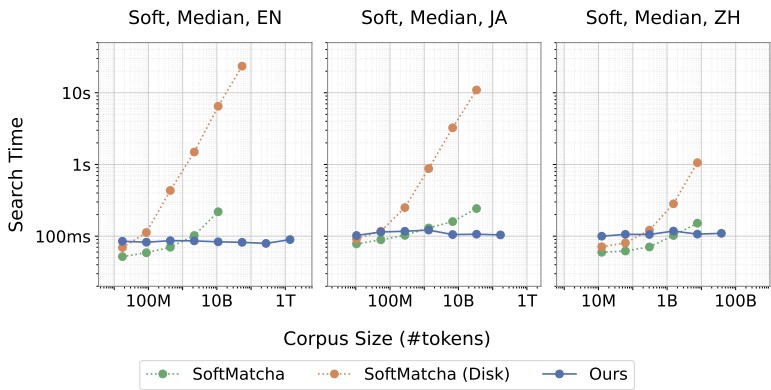

*Figure 11.* The median latency of exact match for EN (FineWeb-Edu, 1.4T tokens) dataset. infini-gram mini timed out during index construction for larger corpora, and an error occurred for smaller corpora.

*Figure 12.* The median latency of soft search for EN (FineWeb-Edu, 1.4T tokens), JA (C4 Japanese, 169B tokens), and ZH (C4 Chinese, 38.3B tokens) datasets. SoftMatcha hit a memory limit, timed out during index construction, or encountered errors for larger corpora.

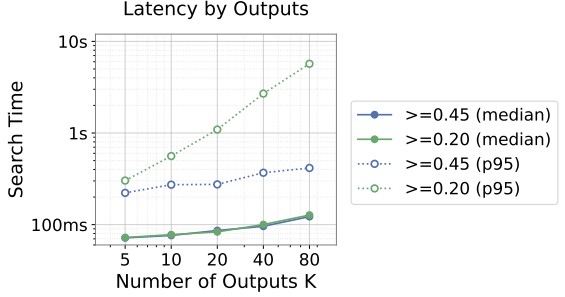

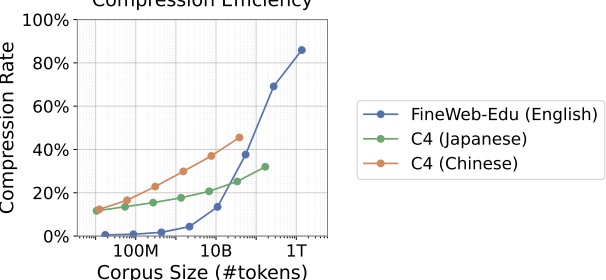

*Figure 13.* Latency when outputting top $K$ patterns with different similarity thresholds ($\alpha = 0.45$ or $0.20$), for FineWeb-Edu dataset.

*Figure 14.* The compression rate of array $X$ in our fast disk-aware suffix array, for various languages and corpus sizes.

*Table 15.* The median and p95 latencies (ms) for real-world data, compared with Gemini-generated queries.

| Dataset | median | p95 |
|---|---|---|
| ORCAS (Craswell et al., 2020) | 54.60 | 433.50 |
| Random FineWeb-Edu Substring | 69.97 | 407.65 |
| Gemini Queries | 66.35 | 187.30 |

*Table 16.* The quality comparison between methods using LLM-as-a-judge for both input orders, where (a) is SoftMatcha, (b) is our method with plain minimum similarity, and (c) is our method with smooth minimum similarity. The numbers are written in the following format: [the number of wins for other than (c)]–[the number of wins for (c)]. The ties ('=') are not counted in both numbers.

| | (c) is first | (c) is second | Win Rate of (c) |
|---|---|---|---|
| (a) and (c) | 127–147 | 85–189 | 61.3% |
| (b) and (c) | 40–222 | 72–191 | 78.7% |

SoftMatcha and Ours (a vs c) and 279 for the comparison between plain minimum and smooth minimum (b vs c). In addition, since LLMs are biased by order (i.e., the latter method B is preferred more regardless of the quality), we conducted experiments for both orderings.

The results are shown in Table 16. For both comparisons, (c) our method with smooth minimum outperforms the other method. However, in terms of the winning rate, the insertion has a negative effect; indeed, the winning rate for (b) was lower than that for (a). We believe that one of the reasons is that some *typo patterns*[15] appear in the top-$K$ outputs if we allow insertions, and this does not mean the insertion lowers the quality of the output.

### I.2. Information Retrieval

Second, as a practical case study, we evaluated the effectiveness of our method for information retrieval tasks. For information retrieval tasks, BM25 (Robertson & Zaragoza, 2009) is a well-known and strong baseline that computes the *similarity score* between the search query and each document using term frequency (TF) and the inverted document frequency (IDF).

SoftMatcha (Deguchi et al., 2025) extended the idea of BM25 by adding a partial score for similar terms. For example, if the query is "flight route" and the term "airline route" is included in a document, we add some partial score to this document[16]. With this expansion, the precision and recall

---

[15] The patterns include some grammar mistakes. For example, when the query is "these are pens", the pattern "these are a pens" should have high similarity, but has a typo (grammar mistake).

[16] Since the similarity between "flight route" and "airline route" is 61.8%, we add a partial score which is $f(0.618)$x compared to the complete-match case for some increasing function $f$.

*Table 17.* The retrieval performance for TREC-COVID and SciFact datasets. P and R indicate precision and recall.

| Method | TREC-COVID | | SciFact | |
|---|---|---|---|---|
| | P@20 | R@1000 | P@1 | R@5 |
| BM25 | 33.8 | 14.9 | **42.7** | 58.1 |
| SoftMatcha | 35.4 | 17.7 | 42.3 | 60.5 |
| Ours | **36.0** | **18.0** | **42.7** | **60.7** |

for information retrieval tasks increased. Now, our interest is whether applying our method instead of SoftMatcha is effective or not. This means that we also add partial scores for patterns with insertions/deletions, and for the similarity function between two patterns, we use the smooth minimum instead of the plain minimum.

**Setup.** As datasets, we used TREC-COVID (Roberts et al., 2021) (171k documents and 50 queries) and SciFact (Wadden et al., 2020) (5.2k documents and 300 queries). Since the queries in the original datasets are long sentences and are not suitable for queries in BM25, SoftMatcha, and Ours, we manually prepared *pattern queries* for this experiment. For example, if the query is "The temperature in Tokyo is 43 degrees", we transform it into pattern queries such as ["temperature", "Tokyo", "43 degrees"].

In each algorithm, we used $\alpha = 0.45$ and $K = 20$ as thresholds for SoftMatcha and Ours. That is, we do not add partial scores for pattern occurrences with similarity less than 0.45 or below rank #21. To evaluate the retrieval performance, we used precision and recall following previous work (Bendersky et al., 2020; Deguchi et al., 2025).

**Results.** The results are shown in Table 17. For both datasets, our method achieved the best performance, which is even better than SoftMatcha. This means that our method is also effective in information retrieval tasks.

### I.3. Paraphrases

Third, for another practical case study, we evaluated the effectiveness of our method in paraphrase detection tasks. More precisely, we consider the task that inputs a *paraphrased* random sentence in a set of documents $S$, and then investigates the document from which the paraphrased text originates. This is useful in plagiarism checks, especially when the plagiarist slightly alters the original text to avoid detection.

**Setup.** For the set of documents $S$, we used 10k random documents from Fineweb-Edu. For the input sentence, we first randomly extracted a sentence between 20 and 50 words from $S$, and paraphrased it by GPT-5.4 with the following prompt:

You are an expert copyeditor. Please rewrite the

*Table 18.* The performance of each method on the paraphrase detection task, expressed as percentages.

| Method | Precision@1 |
| --- | --- |
| infini-gram | 65.5 |
| SoftMatcha | 74.5 |
| Ours | **84.3** |
| BM25 | 82.8 |
| Ours+IDF | **85.8** |

following text to avoid direct plagiarism. Change the vocabulary and phrasing to be unique, but keep the exact original meaning, details, and logical flow intact. Output ONLY the rewritten text without any introductory words.

**Evaluation Metrics.** For the evaluation metrics, we selected Precision@1. It means that just like §I.2, we first measure *similarity score* for each document, and the score is 100% if and only if the document with the highest similarity score exactly matches the original document, and 0% otherwise. We tested 400 queries and calculated the average score.

**Methods and Baselines.** We compared the performance of BM25, infini-gram, SoftMatcha, and Ours. For Ours, we also tested the version that considers IDF when calculating the similarity score (we denote it as Ours+IDF). This means that we compare 5 methods in total.

**Similarity Score.** To calculate the similarity score, we need to transform the paraphrased sentence into pattern queries. For BM25, we take each word as a query, which is standard. For other methods, we take each substring of 4 words as a query[17]. For hyperparameters, we selected $\alpha = 0.35$ and $K = 20$ for SoftMatcha and Ours[18]. For infini-gram, we selected $\alpha = 1.00$ because it only supports exact match.

**Results.** The results are shown in Table 18. Our method outperforms all other methods, including BM25, even without considering IDF. In particular, there is a 9.8-point difference between SoftMatcha and Ours, indicating that insertion/deletion and smooth minimum are effective for paraphrase detection tasks.

### I.4. Sensitivity of Search Quality to $\beta$ Value

Fourth, we tested how the output quality changes as $\beta$ varies. Note that our similarity function in §3 generalizes the plain minimum (used in SoftMatcha) and the simple summation. Indeed, $\beta \to \infty$ corresponds to the plain minimum, and

---

[17] That is, when the sentence length is $\ell$, the list of pattern queries has the size $\ell - 3$.

[18] Although $\alpha$ is small, most search queries finish within 200 ms due to the low $K$ (see Figure 13).

---

$\beta \approx 1$ corresponds to the simple summation.

First, Table 19 shows the search results for the query "paris to beijing" on an 8.6B subset of FineWeb-Edu, for 4 values of $\beta$: $\{1.001, 10, 10^4, 10^{20}\}$.

- When $\beta$ is small, multiple small deviations are cumulatively penalized. Therefore, the ranks of similar patterns in which two or more words are different tend to be lower. For example, the rank of "france to china", which we believe should be higher, was only #9 when $\beta = 10$, and #28 when $\beta = 1.001$. In addition, patterns in which only one word differs tend to rank higher. For example, "paris . beijing" ranks #7 when $\beta \leq 10$, even though the second words "to" and "." are not very similar.

- When $\beta$ is close to infinity, only the worst-matching word determines similarity, and all other positions are ignored. Therefore, the similarity between "london to beijing" (#9, 57.8%) and "london to china" (#10, 57.8%) is identical, although the third word differs only in the latter pattern. In addition, unrelated patterns like "french wanted china" (#6) are likely to get a higher rank.

- However, when $\beta = 10^4$, both problems are resolved and lead to the best quality.

We also tested many other examples and observed a similar trend, indicating that $\beta = 10^4$ is the best choice.

- When $\beta = 1.001$ (summation), completely unrelated patterns with some exact-matched words received higher ranks. For "historical data", unrelated patterns such as "examples data" (#37) and "historical suggest" (#47) received higher ranks (the ranks were only #79 and #96 for the $\beta = 10^4$ case). In addition, for "shortest flight route", the pattern "longest airline routes", which we believe should be similar, ranks below #100 when $\beta = 1.001$, although it ranks #5 when $\beta = 10^4$.

- When $\beta = 10^{20}$ (plain min.), if the worst-matching word is the same, the scores are nearly identical. For example, for "olympics gold medalist", "olympics gold medal" (65.4%) and "olympic silver medal" (65.4%) receive nearly identical scores despite very different semantic changes (65.4% and 63.2% for the $\beta = 10^4$ case). In addition, for "homemade bombs" and when $\beta = 10^{20}$, the top-20 results are dominated by patterns like "bombs exploded" (56.1%, #16) and "bombs detonated" (56.0%, #19), where the first word "homemade" is entirely lost; the minimum is determined solely by the second word.

*Table 19.* The top-10 search results when the search query is "paris to beijing", for various $\beta$ values. Note that our default setting in experiments is $\beta = 10^4$.

| | $\beta = 1.001$ | | $\beta = 10$ | | $\beta = 10^4$ | | $\beta = 10^{20}$ | |
|---|---|---|---|---|---|---|---|---|
| # | Pattern | Sim. | Pattern | Sim. | Pattern | Sim. | Pattern | Sim. |
| 1 | paris to beijing | 100.0% | paris to beijing | 100.0% | paris to beijing | 100.0% | paris to beijing | 100.0% |
| 2 | paris to chinese | 69.1% | paris to chinese | 69.1% | paris to chinese | 69.1% | paris to chinese | 69.1% |
| 3 | paris to shanghai | 64.7% | paris to shanghai | 64.7% | paris to shanghai | 64.7% | france to china | 65.8% |
| 4 | brussels to beijing | 59.1% | brussels to beijing | 59.1% | france to china | 62.9% | paris to shanghai | 64.7% |
| 5 | paris to seoul | 59.1% | paris to seoul | 59.1% | brussels to beijing | 59.1% | french to chinese | 59.8% |
| 6 | london to beijing | 57.8% | london to beijing | 57.8% | paris to seoul | 59.1% | french wanted china | 59.2% |
| 7 | paris . beijing | 57.3% | paris . beijing | 57.3% | london to beijing | 57.8% | brussels to beijing | 59.1% |
| 8 | paris to taiwan | 56.7% | paris to taiwan | 56.7% | paris . beijing | 57.3% | paris to seoul | 59.1% |
| 9 | paris to hong | 53.6% | france to china | 54.0% | paris to taiwan | 56.7% | london to beijing | 57.8% |
| 10 | paris and beijing | 53.1% | paris to hong | 53.6% | london to china | 56.3% | london to china | 57.8% |

- However, when $\beta = 10^4$, similar results are included without excess or deficiency. For "homemade bombs", results correctly include "home-made bombs" (57.2%, #8) and "handmade bomb" (52.9%, #18), all semantically relevant while excluding unrelated patterns that appear at extreme $\beta$ values. For "stock price change", the top-20 includes "market price change" (67.4%, #8), "share price change" (61.0%, #15), and inflectional variants, providing a coherent set.

## J. Subword Tokenizers

In recent large language models (LLMs), subword tokenizers such as GPT-2 and LLaMA are often used; these differ significantly from word-level tokenizers such as GloVe. Accordingly, we also conducted experiments using `meta-llama/Llama-2-7b-hf` model (Touvron et al., 2023), a subword tokenizer, on FineWeb-Edu/sample-10BT dataset (8.6B tokens with the GloVe tokenizer, 11.2B tokens with the LLaMA-2 tokenizer).

**Qualitative Evaluation.** Table 20 shows the results when we searched for "gold medal" and "chocolate". For the first query, the results are as expected, although the similarity is lower than that of the GloVe tokenizer, and many case-substitutions are displayed because the LLaMA-2 tokenizer is case-sensitive. For the second query, however, the output includes many spurious matches. This is because, in LLaMA-2, the word "chocolate" is tokenized as ["ch", "oc", "olate"], and the top-$K$ patterns are determined by the cosine similarities of individual tokens. However, more than 83% of word occurrences in this corpus were not split into multiple tokens, and cases such as "chocolate" are rare. Therefore, we expect our method to work reasonably well (especially after aligning uppercase and lowercase letters).

**Latency.** When we set the similarity threshold to 0.1, we achieved a median latency of 57.86 ms and a p95 latency of 145.90 ms for outputting the top $K = 20$ patterns, which is comparable to that of the word-level tokenizer. Of the 400

*Table 20.* Example search results on FineWeb-Edu/sample-10BT using LLaMA-2 tokenizer.

| | Query: gold medal | | | Query: chocolate | |
|---|---|---|---|---|---|
| # | Pattern | Sim. | # | Pattern | Sim. |
| 1 | _gold _medal | 100% | 1 | _ch oc olate | 100% |
| 2 | _Gold _medal | 53.7% | 2 | _Ch oc olate | 41.3% |
| 3 | _gold _Medal | 47.3% | 3 | Ch oc olate | 26.0% |
| 4 | _Gold _Medal | 42.5% | ⋮ | | |
| 5 | _silver _medal | 27.3% | 8 | _Ch oc ocoa | 18.4% |
| ⋮ | | | ⋮ | | |
| 20 | _steel _medal | 15.2% | 20 | _C oc olate | 11.4% |

cases, 8 failed because the number of tokens exceeded the upper limit of 12 (see Appendix L). Of the remaining 392 cases, the program returned the maximum of 20 patterns in 199 cases (51%) and at least 1 pattern in 341 cases (87%).

## K. Details of Dynamic Corpus-Aware Pruning in §3.3

As explained in §3.3, we introduced two additional techniques, in addition to iterative pruning.

### K.1. $k$-gram Pruning: Caching 2-gram and 3-gram Existence

To reduce the number of corpus-existence checks that require disk access, we store precomputed 2-gram and 3-gram existence tables in RAM. Specifically, we record (1) all 2-grams both of whose constituent words rank among the top 100K most frequent words and (2) all 3-grams whose three constituent words have frequency ranks that sum to 10K or less. When calculating $R_1, R_2, \dots$ (see §3.3), if a 2-gram or 3-gram meets the frequency-rank conditions above (and thus would appear in the table if it existed in the corpus), we skip the corpus lookup when it is absent from the tables.

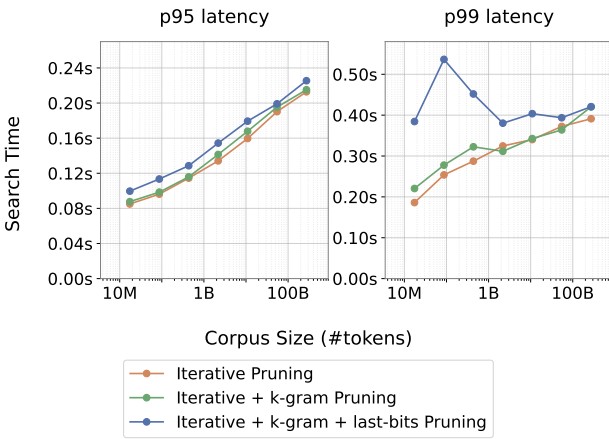

*Figure 15.* The improvement on p95 and p99 latencies when we applied pruning for FineWeb-Edu dataset (n = 2,000).

### K.2. Last-Bits Pruning: For Patterns with Low Frequencies

Suppose that in Algorithm 1, the current pattern $w \in R_{i-1}$ appears only once in the corpus. In this case, it is inefficient to search for $w \cdot w'$ for all $w' \in \mathcal{V}^*$ with similarity $\alpha$ or more. Instead, it is better to look at the corpus directly and find the next tokens. We applied this method when the number of occurrences (after run-length compression of array $X$ in Appendix L) is 50 or less.

### K.3. Effects on Pruning

Next, we show the effectiveness of each pruning in terms of latency. Figure 15 shows the p95 and p99 latencies (1) when we only applied iterative pruning; (2) when we applied iterative pruning and $k$-gram pruning; and (3) when we applied all of iterative pruning, $k$-gram pruning, and last-bits pruning. The $k$-gram and last-bits prunings work well in smaller corpora; the p99 latencies improved by approximately 2x for corpora smaller than 1B tokens. However, the improvement in latency was not as strong as the improvement in the number of searches (see §5.4), mainly due to the page cache. Note that in this experiment, we specially tested on 2,000 Gemini queries instead of the normal 400 queries. This is because, to evaluate p99 latencies, 400 samples are too few.

## L. Details of the Fast Disk-Aware Suffix Array in §3.4

As explained in §3.4, we also improve the suffix array to further improve the efficiency of exact lookup. The core ideas are (1) to exploit the fact that queries are usually short (10 tokens or fewer) and (2) to store some indices in RAM to reduce the number of random disk accesses.

*Table 21.* The estimated value of $\mathbf{E}[|R'_i|]/\mathbf{E}[|R_{i-1}|]$ with a corpus of an 8.6B subset of FineWeb-Edu, when $\alpha = 0.50$.

| $i$ | 1 | 2 | 3 | 4 | 5 |
|---|---|---|---|---|---|
| Without Insert/Delete | 16.9 | 8.1 | 5.7 | 5.5 | 8.8 |
| With Insert/Delete | 16.9 | 26.2 | 10.2 | 13.6 | 34.5 |

Let $L$ be the maximum query length (in our experiments, we used $L = 12$). The first step is to record a sorted array of all contiguous $L$ patterns in the corpus (i.e., the array has length $|\mathcal{C}| - L + 1$), instead of the suffix array; we denote this array by $X$. By binary search on $X$, we can count the number of occurrences of the query string $\bar{q}$ in the corpus. However, it still requires $O(\log |\mathcal{C}|)$ random disk accesses. We further reduce the number of accesses using a two-level indexing strategy, similar to Google's BigTable (Chang et al., 2008), described below.

Let $B$ be a constant.[19] In addition to $X$, we construct a sparse array $Y = [X_0, X_B, X_{2B}, \ldots]$, where $X_i$ is the $i$-th element of the array $X$. This array is stored in RAM. When we search for the position of an occurrence of query $\bar{q}$, we first find the approximate position (i.e., such $i$ that $\bar{q}$ occurs in $Z = [X_{iB}, X_{iB+1}, \ldots, X_{(i+1)B-1}]$ if it exists in $X$) using $Y$, and then find the exact position in $X$ using $Z$. Surprisingly, only a single random disk access is required, significantly fewer than the $O(\log |\mathcal{C}|)$ accesses required by a standard suffix array if $B$ is set appropriately so that the $B/2$ entries are smaller than the page size of operating systems. This is because the range of the array $X$ examined (in the second step of binary search) is at most $B/2$ entries.

**Compression.** This method requires substantial disk space. Let $V$ be the vocabulary size; then the size of $X$ is $|\mathcal{C}|\lceil(L\lceil\log_2 V\rceil)/8\rceil$ bytes (e.g., $29|\mathcal{C}|$ bytes when $L = 12$ and $V = 400K$). To address this problem, we applied run-length compression to $X$, exploiting the many duplicate sequences that arise primarily from citations.

## M. Empirical Validation for §4

In this section, we empirically validate assumptions of our theoretical analysis in §4.

### M.1. Validation of Assumption 1

To validate Assumption 1, we measured $\mathbf{E}[|R'_i|]/\mathbf{E}[|R_{i-1}|]$, which is, the mean value of $|R'_i|$ divided by the mean value of $|R_{i-1}|$. We set $\alpha = 0.50$, and for corpus and queries, we used an 8.6B-subset of FineWeb-Edu and 400 Gemini-generated queries.

**Validation.** Table 21 shows the estimated value of $\mathbf{E}[|R'_i|]/\mathbf{E}[|R_{i-1}|]$ with and without insertions/deletions.

---

[19] In our experiments, we use $B$ between 128 and 256.

*Table 22.* The log-shifted geometric means of $|R_i|$ and $|S_i|$, where $S_i$ is the set of patterns (including those not in the corpus) whose similarity to the $i$-token prefix $q_1 q_2 \cdots q_i$ of the query is $\alpha$ or more, and $R_i$ is the set of patterns in $S_i$ that actually occur in the corpus. The table also shows $\rho_i$, the corpus occurrence proportion among all possible $i$-token patterns.

| $i$ | | $\alpha = 0.70$ | | | $\alpha = 0.60$ | | | $\alpha = 0.50$ | | $\rho_i$ |
| | $|R_i|$ | $|S_i|$ | $|R_i|/|S_i|$ | $|R_i|$ | $|S_i|$ | $|R_i|/|S_i|$ | $|R_i|$ | $|S_i|$ | $|R_i|/|S_i|$ | |
|---|---|---|---|---|---|---|---|---|---|---|
| **Without Insert/Delete** 1 | 1.8 | 1.8 | 0.998 | 4.4 | 4.4 | 0.998 | 16.8 | 16.9 | 0.998 | 0.957 |
| 2 | 2.8 | 2.9 | 0.971 | 10.4 | 11.5 | 0.904 | 108.3 | 136.7 | 0.792 | 0.001 |
| 3 | 3.0 | 4.2 | 0.714 | 11.8 | 27.6 | 0.428 | 103.1 | 727.8 | 0.142 | 0.000 |
| 4 | 2.5 | 5.7 | 0.435 | 8.1 | 56.2 | 0.144 | 44.6 | 2,681.3 | 0.017 | 0.000 |
| 5 | 2.3 | 8.1 | 0.288 | 6.4 | 118.3 | 0.054 | 29.4 | 10,665.1 | 0.003 | 0.000 |
| **With Insert/Delete** 1 | 1.9 | 1.9 | 0.998 | 4.4 | 4.4 | 0.998 | 16.9 | 16.9 | 0.998 | 0.957 |
| 2 | 6.9 | 14.4 | 0.483 | 21.0 | 60.3 | 0.348 | 149.4 | 445.6 | 0.335 | 0.001 |
| 3 | 7.3 | 33.7 | 0.216 | 21.2 | 198.1 | 0.107 | 136.6 | 2,622.9 | 0.052 | 0.000 |
| 4 | 6.0 | 78.0 | 0.077 | 14.9 | 670.9 | 0.022 | 64.4 | 12,513.0 | 0.005 | 0.000 |
| 5 | 5.9 | 267.7 | 0.022 | 13.7 | 2,980.3 | 0.005 | 48.7 | 74,976.4 | 0.001 | 0.000 |

Even with insertions and deletions, we believe that $A \leq 40$ is enough.

## M.2. Validation of Assumption 2

To validate Assumption 2 with respect to Remark 1, we measured the log-shifted mean[20] of $|R_i|$ and $|S_i|$ when $K = 20$. For a corpus, we used an 8.6B-token subset of FineWeb-Edu. For queries, we used 400 Gemini-generated queries.

**Validation.** Table 22 shows the log-shifted geometric mean of $|R_i|$ and $|S_i|$ for each $i$, with and without insertions and deletions. In this table, $\rho_i$ described in Remark 1, which is the proportion of the $i$-grams in the corpus among all possible $i$-token patterns, is also described.

Since the vocabulary size is large, $|R_i|$ begins to deviate from $\rho |S_i|$ at $i = 2$ (Remark 1). However, Assumption 2 still holds at $(a, r) = (300, 0.7)$ for the case with insertions/deletions and $\alpha = 0.50$.

The numbers described in the table are just a log-shifted geometric mean over the queries; they do not mean that $r \approx 0.7$ for *all* queries. In some cases, $|R_i|$ does not converge, that is, $r \geq 1$.

For example, when the search query is "play an important role in the development of", the values $[|R_1|, \ldots, |R_8|]$ are $[37, 262, 277, 150, 551, 1040, 193, 230]$, and the trendline is $r = 1.029$ even if we exclude $|R_1|$. Note however that the exponential explosion of $R_i$ is much milder than that of $S_i$, which accounts for the efficiency of our algorithm.

## N. Details of the Theoretical Analysis

This section provides the details of the theoretical analysis in §4. We give the full proofs of Theorem 1 and Theorem 2 (§N.1 and §N.2). We also discuss how our analysis relates to

---

[20] We used log-shifted geometric means because a simple mean can be affected too much by a single outlier.

the adaptive-threshold procedure and the additional pruning heuristics used in our implementation (§N.3).

## N.1. Proof of Theorem 1

Now we prove Theorem 1, the sublinearity of $\mathbf{E}[Total]$ with respect to $|\mathcal{C}|$. The proof uses the standard Zipf-to-Heaps calculation: if the $n$-gram distribution follows Zipf's law with exponent $\delta > 1$, then the number of distinct observed $n$-grams grows sublinearly in corpus size (Heaps' law).

**Lemma 1** (Zipf's law to Heaps' law). *Assume that the frequency distribution of $n$-grams satisfies Zipf's law with $\delta > 1$ for each $n$; more precisely, assume that the $k$-th most frequent $n$-gram occurs in the corpus approximately independently at each position with a probability $p_{n,k}$ such that $p_{n,k} \leq c_n k^{-\delta}$ for some constant $c_n > 0$ for each $n$ and $k$. Then the expected number of distinct $n$-grams observed in $\mathcal{C}$ is $\mathbf{E}\big[|\bar{\mathcal{C}}_n|\big] \approx O\big(|\mathcal{C}|^{1/\delta}\big)$, where the hidden constant may depend on $n$, $c_n$ and $\delta$, but not on $|\mathcal{C}|$.*

*Proof.* Let $I_{n,k}$ be the indicator variable for the event that the $k$-th most frequent $n$-gram occurs at least once in $\mathcal{C}$. Then $|\bar{\mathcal{C}}_n| = \sum_{k \geq 1} I_{n,k}$. By linearity of expectation, $\mathbf{E}\big[|\bar{\mathcal{C}}_n|\big] = \sum_{k \geq 1} \Pr[I_{n,k} = 1]$. By the union bound, letting $N_n \triangleq |\mathcal{C}| - n + 1$ be the number of $n$-gram positions in $\mathcal{C}$, $\Pr[I_{n,k} = 1] \lesssim \min(1, N_n p_{n,k}) \leq \min(1, |\mathcal{C}| c_n k^{-\delta})$. Therefore, $\mathbf{E}\big[|\bar{\mathcal{C}}_n|\big] \lesssim \sum_{k \geq 1} \min(1, |\mathcal{C}| c_n k^{-\delta})$.

Let $K \triangleq \lfloor (|\mathcal{C}| c_n)^{1/\delta} \rfloor$. Note that $\min(1, |\mathcal{C}| c_n k^{-\delta}) = 1$ for $k \leq K$ and $\min(1, |\mathcal{C}| c_n k^{-\delta}) = |\mathcal{C}| c_n k^{-\delta}$ for $k > K$. Using $\delta > 1$ and $K = O\big(|\mathcal{C}|^{1/\delta}\big)$, we have

$$
\begin{aligned}
\sum_{k > K} |\mathcal{C}| c_n k^{-\delta} &\leq |\mathcal{C}| c_n \int_K^\infty x^{-\delta} \, dx \\
&= \frac{|\mathcal{C}| c_n}{\delta - 1} K^{1-\delta} \\
&= O\Big(|\mathcal{C}| \cdot (|\mathcal{C}|^{1/\delta})^{1-\delta}\Big)
\end{aligned}
$$

$$= O\left(|\mathcal{C}|^{1/\delta}\right).$$

We also have $\sum_{k \le K} 1 = O(K) = O\left(|\mathcal{C}|^{1/\delta}\right)$. Therefore, combining the two parts $k \le K$ and $k > K$, we obtain

$$\mathbf{E}[|\bar{\mathcal{C}}_n|] \approx O\left(|\mathcal{C}|^{1/\delta}\right) + O\left(|\mathcal{C}|^{1/\delta}\right) = O\left(|\mathcal{C}|^{1/\delta}\right).$$

$\square$

We also introduce the following utility lemma:

**Lemma 2.** *[Bound Total with $|R_i|$] Under Assumption 1,*
$\mathbf{E}[Total] \le A \sum_{i=0}^{m-1} \mathbf{E}[|R_i|]$.

*Proof.* By the fact $Total = \sum_{i=1}^{m} |R_i'|$ and Assumption 1, we have

$$
\begin{aligned}
\mathbf{E}[Total] &= \sum_{i=1}^{m} \mathbf{E}[|R_i'|] \le \sum_{i=1}^{m} A\,\mathbf{E}[|R_{i-1}|] \\
&= A \sum_{i=0}^{m-1} \mathbf{E}[|R_i|].
\end{aligned}
$$

The first equation holds by linearity of expectation. $\square$

*Proof of Theorem 1.* By Lemma 1, fed with the first assumption, and the second assumption of the theorem, we obtain

$$\mathbf{E}[|R_i|] = O\big(\mathbf{E}[|\bar{\mathcal{C}}_i|]\big) \approx O\left(|\mathcal{C}|^{1/\delta}\right)$$

for each fixed $i \le m$. Therefore, by Lemma 2, we have

$$
\begin{aligned}
\mathbf{E}[Total] &\le A \sum_{i=0}^{m-1} \mathbf{E}[|R_i|] \\
&\approx A \sum_{i=0}^{m-1} O\left(|\mathcal{C}|^{1/\delta}\right) = O\left(|\mathcal{C}|^{1/\delta}\right).
\end{aligned}
$$

The last equation holds since $m$ is finitely fixed. $\square$

### N.2. Proof of Theorem 2

Theorem 2 can be proved by straightforward calculation, under Assumption 2.

*Proof of Theorem 2.* By Lemma 2 and Assumption 2, under $r \ne 1$, we have

$$
\begin{aligned}
\mathbf{E}[Total] &\le A \sum_{i=0}^{m-1} \mathbf{E}[|R_i|] \\
&\le A \sum_{i=0}^{m-1} a r^i = Aa(1 - r^m)/(1 - r).
\end{aligned}
$$

$\square$

### N.3. Relation to Adaptive Thresholding and Additional Pruning

The analysis above is stated for a single fixed threshold $\alpha$. The implemented system uses adaptive thresholding: it runs the same core search for a finite sequence of thresholds $\alpha_1 > \alpha_2 > \cdots > \alpha_N$ until enough results are found or a floor threshold is reached. If $N$ is bounded and the assumptions above hold uniformly for all thresholds above the floor, then the adaptive version incurs only a constant-factor overhead:

$$Total_{\text{adapt}} \le \sum_{j=1}^{N} Total(\alpha_j).$$

Thus the $O(1)$ query-length bound is preserved under adaptive thresholding, provided the constants in the assumptions can be chosen uniformly over the thresholds considered by the adaptive procedure.

The additional pruning techniques used in the implementation, such as $k$-gram pruning and last-bits pruning, only remove exact lookups that the core algorithm would otherwise perform. Therefore they cannot increase $Total$, and all upper bounds above continue to apply.

## O. Details of Contamination Detection in §6

The list of 7 benchmarks we used is shown in Table 27. In addition, Tables 23 to 26 show 36 examples of benchmark contamination that were flagged as dirty only by soft search, along with manual verification results (for a discussion of benchmark contamination, see §6).

*Table 23.* List of 36 examples of benchmark contamination which were flagged as dirty only by soft search, along with manual verification results (example 1–10). The bolded parts indicate the soft matches, and that of the Problem section spans 10 tokens. The **blue**, **magenta**, and ~~gray~~ parts indicate substitution, insertion, and deletion.

| # | Dataset | Problem | One of the most similar texts in corpus | Result |
|---|---------|---------|------------------------------------------|--------|
| 1 | MMLU | The difference between dc and **ac in electric circuits is that in dc, charges** flow | ... Please click on the link below to read Our Vision Statement. The difference between DC and **AC** ~~[in]~~ **electric circuits is that in DC, charges** flow in one direction ... | Semantic contamination |
| 2 | MMLU | This is a part of the communication process where the sender selects a combination of **appropriate words, pictures, symbols, and music to** represent a message to be transmitted: | ... Question 12 This is a part of the communication process where the sender selects a combination of **appropriate words, pictures, symbols** ~~[,]~~ **and music to** represent a message to be transmitted: a) Encoding. b) Decoding. c) Transfer. d) Feedback. ... | Semantic contamination |
| 3 | MMLU | **Which of the following conditions is caused by a trinucleotide** (triplet) repeat expansion? | ... 7. **Which of the following conditions is caused by** ~~[a]~~ **trinucleotide** (triplet) repeat expansion? a) Cystic fibrosisb) Duchenne muscular Dystrophyc) Huntington diseased) ... | Semantic Contamination |
| 4 | MMLU | The two most common causes **of foodborne illness in the United States and Europe are**: | ... Foodborne pathogenic bacteria such as, pathogenic E. coli, Listeria and Salmonella and enteric viruses such as human norovirus, hepatitis A and E viruses are the most common causes **of foodborne illness in the United States and Europe** ~~[are]~~. However, bacterial and viral microorganisms differ ... | Semantic Contamination |
| 5 | MMLU | There are two games involving flipping a fair coin. In the first game you win a prize if you can throw between 45% and 55% heads. In the second game you win if you can **throw more than 80% heads. For each game** would you rather flip the coin 30 times or 300 times? | ... 3) Whiteboard Review There are two games involving flipping a fair coin. In the first game, you win a prize if you can throw between 45% and 55% heads; in the second game, you win if you can **throw more than 60% heads. For each game**, would you rather flip the coin 30 times or 300 times? A.30 times for each game. B.300 times for each game ... | Template Leakage |
| 6 | MMLU | **What factors increase the likelihood of burnout?** | ... **What Factors Increase the Risk of Burnout?** The risk of burnout is different for everyone, and it depends | Semantic Contamination ... |
| 7 | MMLU | **Electrical stimulation of the brain for the treatment of chronic** pain | ... Forty-eight patients underwent **electrical stimulation of the brain for** ~~[the]~~ **treatment of chronic** pain between 1978 and 1983. Average pain duration prior to treatment was 4.5 years. ... | False Positive |
| 8 | MMLU | The ethics of conducting archaeological research on sites with human remains of indigenous North American ancestors is addressed by **the Native American Graves Protection and Repatriation Act. Problems** often arise when: | ... to see that they were properly buried in accordance with **the Native American Graves Protection and Repatriation Act. The trouble** is that Kennewick Man does not appear to have ... | Semantic Contamination |
| 9 | MMLU | **What is 123 / 8?** | ... How to Find the Value of 163/4 **What Is 163/4?** 163/4 is an algebraic expression. It is a fraction, which means it is composed of two parts, a numerator and a denominator. ... | Template Leakage |
| 10 | MMLU | **What do you call three consecutive strikes in bowling?** | ... a. - Virginia limestone b. - white marble c. - sandstone d. - Virginia granite 3. **What do we call three consecutive strikes in bowling ?** ... | Semantic Contamination |

*Table 24.* List of 36 examples of benchmark contamination which were flagged as dirty only by soft search, along with manual verification results (example 11–20). The bolded parts indicate the soft matches, and that of the Problem section spans 10 tokens. The **blue**, **magenta**, and ~~gray~~ parts indicate substitution, insertion, and deletion.

| # | Dataset | Problem | One of the most similar texts in corpus | Result |
|---|---------|---------|------------------------------------------|--------|
| 11 | MMLU | _____________ is a **popular tool used for network analysis in multiprotocol diverse network**. | ... Which of the following is a **popular tool used for network analysis in multiprotocol diverse networks**? ... | Semantic Contamination |
| 12 | MMLU | **In order to go from national income to GDP one must** | ... See All test questions 1. **In order to go from national income to GDP, one must** 2. Which of the following creates the trade-offdepicted by the Phillips curve? ... | Semantic Contamination |
| 13 | MMLU | **Find the GCD of 25 and 55.** | ... Step 1: Firstly, we need to **find the GCD of 30 and 56.** The GCD is the largest number that divides evenly into both 30 and 56. In this case, the GCD of 30 and 56 is 2. Step 2: Dividing both the numerator and the denominator by the GCD will simplify the fraction. ... | False Positive |
| 14 | MMLU-Pro | Suppose the economy is in long-run equilibrium when a temporary expansionary supply shock is felt in the economy. This changes the short-run **Phillips curve the short-run unemployment rate and the long-run unemployment** rate in which of the following ways? SHORT-RUN PHILLIPS CURVE ... | ... 53 . Suppose the economy is in long-run equilibrium when a temporary expansionary supply shock is felt in the economy. This changes the short-run **Phillips curve, the short-run unemployment** rate, and the long-run unemployment rate in which of the following ways? ... | Semantic Contamination |
| 15 | MMLU-Pro | Find the smallest positive integer that leaves a remainder of 2 when divided by 3, a remainder of 3 when divided by 5, **and a remainder of 1 when divided by 7.** | ... Suppose you wished to know, for no apparently useful reason, what number had a remainder of 2 when divided by 3; a remainder of 3 when divided by 4 **and a remainder of 1 when divided by 5.** We'll give the workings in a moment, but for our less patient readers the answer is 71. ... | Template Leakage |
| 16 | MMLU-Pro | If households are more optimistic about **the future how would the consumption function be affected?** | ... 2. If households are more optimistic about **the future, how would the consumption function be affected?** 3. U.S. real GDP most likely falls when ... | Semantic Contamination |
| 17 | MMLU-Pro | Let P be a procedure that for some inputs calls itself (i.e., is recursive). If P is guaranteed to terminate, which of the following statements must be true? I. P has a local **variable. II. P has an execution path where** it does not call itself. III. P either refers to a global variable or has at least one parameter. | ... If P is guaranteed to terminate, which of the following statement(s) must be true? I. P has a local **variable** ~~.~~ **II. P has an execution path where** it does not call itself III. P either refers to a global variable or has at least one parameter (A) I only (B) II only (C) III only (D) II and III only ... | Semantic Contamination |
| 18 | MMLU-Pro | Use the list of numbers: 22, 25, 14, 11, **23, 27, 46. What is the mode**? | ... Use the list of numbers to answer the question below. 22, 25, 14, 11, **23, 27, 46** ~~.~~ **What is the mode**? A. 23 B. 24 C. 35 D. No mode ... | Semantic Contamination |
| 19 | MMLU-Pro | "Natural law is based on the nature of man and on his inward need **of living in society." Who said it?** | ... 12. "Natural law is based on the nature of man and on his inward need **of living in society" . Who said it?** (a) Hugo Grotius Answers: 1.(c) ... | Semantic Contamination |
| 20 | MMLU-Pro | ... "and of their alterations as observed more precisely later on we shall give a description here. Also I measured **the distances between them by means of the telescope**.... | ... and of their alterations as observed more precisely later on we shall give a description here. Also I measured **the distances between them by means of the telescope** ~~.~~, using the method explained before. Moreover I recorded ... | Semantic Contamination |

*Table 25.* List of 36 examples of benchmark contamination which were flagged as dirty only by soft search, along with manual verification results (example 21–29). The bolded parts indicate the soft matches, and that of the Problem section spans 10 tokens. The **blue**, **magenta**, and ~~[gray]~~ parts indicate substitution, insertion, and deletion.

| # | Dataset | Problem | One of the most similar texts in corpus | Result |
|---|---------|---------|----------------------------------------|--------|
| 21 | MMLU-Pro | For matrix A = [[3, 1, 1], [2, 4, 2], **[1, 1, 3]], what** are its eigen values? | ... If we wanted to sort this list by the second element in each list so that the result would be [[4, 0, 1], [2, 1, 3], **[1, 2, 3]], what** function would we write to pass as the key value to the sort() method? ... | False Positive |
| 22 | MMLU-Pro | ... German history reached its turning-point and failed to turn. This was the fateful essence of 1848. A. J. P. **Taylor, The Course of German History, 1945 The** subject of Taylor's analysis in this quotation is | ... This was the fateful essence of 1848. A. J. P. **Taylor, The Course of German History, 1945** ~~[The]~~ 21. What is the subject of Taylor's analysis? A. The Industrial Revolution in the context of German history B. The failure of the revolutions of 1848 ... | Semantic Contamination |
| 23 | MMLU-Pro | ... divided into so many sovereign and independent communities, under a conviction of the absolute necessity of uniting all our councils and all our strength, to maintain and defend **our common liberties⋯." Journals of the Continental** Congress, 1777 The most notable achievement of the United States under the Articles of Confederation was | ... under a conviction of the absolute necessity of uniting all our councils and all our strength, to maintain and defend **our common liberties⋯** ~~[-]~~**" Journals of the Continental** Congress ... | Semantic Contamination |
| 24 | GSM8K | A football team played 22 games. They won **8 more than they lost. How many did they** win? | ... I might say, "In a season of 74 games, the basketball team lost some games, but they won **28 more than they lost. How many did they** win?" Then, assign variables. If possible, assign a variable to the value the problem asks you to find. Here, let: W = number of wins ... | Template Leakage |
| 25 | MATH | How many times does the digit 8 appear in **the list of all integers from 1 to 1000?** | ... How many times does the digit 9 appear in **the list of all integers from 1 to 500?** (The number $ 99 $, for example, is counted twice, because ... | Template Leakage |
| 26 | MATH | The sum of two numbers is 25 and **their difference is 11. What is the larger of** the two numbers? | ... The sum of two numbers is 139 while **their difference is 19. What is the larger of** two numbers? Let the two numbers be x and y. x + y = 139 ... | Template Leakage |
| 27 | MATH | The sum of two numbers is 40 and **their difference is 12. What is their product?** | ... increased by 9 times a number. What is the number? - The sum of two numbers is 10 and **their difference is 4. What is their product?** ... | Template Leakage |
| 28 | MATH | Let A = 1, B = 2, C = 3, ..., Z = 26. The product value of a word is equal to the product of the values of its letters. For example, CAB has a product value **of 3 $\times$ 1 $\times$** 2 = 6. What common English word has a product value of 715? It does not have to be of length 3. | ... such as 3 were really helpful because they only had one option **of 1 $\times$ 1 $\times** $ 3. I also realised that the products that had 5 in them would have to have a factor of 5 otherwise it would be impossible to make that product. ... | False Positive |
| 29 | MATH | The equation $y = -16t^2 + 26t **+ 105$ describes the height (in feet)** of a ball tossed up in the air at 26 feet per second from a height of 105 feet above the ground. In how many seconds will the ball hit the ground? Express your answer as a decimal rounded to the nearest tenth. | ... The equation $h = -16t^2 - 24t **+ 160$ describes the height (in feet)** of the ball. In how many seconds will the ball hit the ground? Express your answer as a decimal. ... | Template Leakage |

*Table 26.* List of 36 examples of benchmark contamination which were flagged as dirty only by soft search, along with manual verification results (example 30–36). The bolded parts indicate the soft matches, and that of the Problem section spans 10 tokens. The **blue**, **magenta**, and ~~gray~~ parts indicate substitution, insertion, and deletion.

| # | Dataset | Problem | One of the most similar texts in corpus | Result |
|---|---------|---------|------------------------------------------|--------|
| 30 | MATH | It took Lara five days to read a novel. Each day after the first day, Lara read half as many pages as the day before. **If the novel was 248 pages long, how many** pages did she read on the first day? | ... Each day after the first day, Lara read half as many pages as the day before. **If the novel was 310 pages long, how many** pages did she read on the first day? x + 1/2 x + 1/4 x + 1/8x + 1/16x = 310 31/16 x = 310 x = first day = 160 pages ... | Template Leakage |
| 31 | MATH | If $3x**+5=29$, what is the value** of $x$? | ... By manipulating equations containing variables to find their values. In the equation $x **+ 7 = 19$, what is the value** of $x$? ... | False Positive |
| 32 | MATH | A pyramid has **6 vertices and 6 faces. How many edges** does it have? | ... Example 2: A polyhedron has **6 vertices and 8 faces. How many edges does** it have? We can use formula of euler ⇒ F + V - E = 2 Faces(F) = 8 ... | Template Leakage |
| 33 | ARC-Challenge | **Which form of energy can travel through a vacuum?** | ... Which of these can travel through a vacuum? **Which form of energy Cannot travel through vacuum?** Heat light and sound are all forms of energy. Heat can be transferred by ... | Semantic Contamination |
| 34 | ARC-Challenge | **Which describes a chemical change?** (choices) 'sugar dissolving in tea', 'sunshine warming a sidewalk', 'scissors cutting paper into pieces', 'acid in the stomach digesting food' | ... Q7 7. Chemical change is a change in its color, smell, or odor and the formation of bubbles. **Which describe a chemical change?** ice turning into water crushing a can mixing oil and water ripening of mango ... | False Positive |
| 35 | ARC-Challenge | What is the mass **of a carbon atom that has 6 protons, 7** neutrons, and 6 electrons? | ... Atomic mass unit (u or amu) Unit of measurement for the masses of particles; 1/12 the mass **of a carbon atom that has 6 protons, 6** neutrons, and 6 electrons. ... | False Positive |
| 36 | ARC-Challenge | An atom of tin has an atomic number of 50 and **a mass number of 119. How many protons,** electrons, and neutrons are found in one neutral atom of tin? | ... Source: Read Full Article Xenon has an atomic number of 54. A particular isotope of xenon has **a mass number of 131. How many protons** ~~,~~ and how many neutrons does each atom of that isotope have? ... | Template Leakage |

*Table 27.* List of 7 benchmarks we used for the contamination experiment.

| Genre | Benchmark | URL |
|-------|-----------|-----|
| Knowledge & Reasoning | MMLU (Hendrycks et al., 2020) | `https://huggingface.co/datasets/cais/mmlu` |
| | MMLU-Pro (Wang et al., 2024) | `https://huggingface.co/datasets/TIGER-Lab/MMLU-Pro` |
| | GPQA (Rein et al., 2024) | `https://huggingface.co/datasets/Idavidrein/gpqa` |
| Math | GSM8K (Cobbe et al., 2021) | `https://huggingface.co/datasets/openai/gsm8k` |
| | MATH (Hendrycks et al., 2021) | `https://huggingface.co/datasets/EleutherAI/hendryks_math` |
| Coding | HumanEval (Chen, 2021) | `https://huggingface.co/datasets/openai/openai_humaneval` |
| Commonsense Understanding | ARC-Challenge (Clark et al., 2018) | `https://huggingface.co/datasets/allenai/ai2_arc` |

