# OpenReview forum: "SoftMatcha 2: A Fast and Soft Pattern Matcher for Trillion-Scale Corpora"
_ICML.cc/2026/Conference — ICML 2026 regular_

### Official Review · Reviewer_ht7D · 2026-03-09

**Soundness:** 2
**Presentation:** 1
**Significance:** 3
**Originality:** 2
**Overall Recommendation:** 3
**Confidence:** 4

**Summary:**

The paper develops a search algorithm for exact and "soft" queries over huge ordered text corpora. A query is defined as an ordered sequence of tokens/words. Soft queries are \emph{semantic} variation of the original query, with substituted, inserted, or deleted tokens. These "patterns" are generated by enumerating possible variations of the original query that may appear in the dataset. The proposed algorithm has two main components: (1) a fast disk-based exact lookup procedure built on suffix arrays, and (2) a dynamic pruning procedure that iteratively removes candidate patterns using corpus-occurrence information obtained during search. The paper provides an informal theoretical analysis, along with a numeric evaluation against benchmark methods, with particular emphasis on a very large corpus containing 1.4 trillion tokens. The paper also presents a contamination-detection application showing the benefit of soft queries by identifying additional near-matching benchmark sequences beyond what exact-matching algorithms can find.

**Compliance With Llm Reviewing Policy:**

Affirmed.

**Final Justification:**

My main concerns remain only partially resolved. The added paraphrase-style retrieval result is a useful step toward evaluating semantic search quality, but it is still relatively limited, and I still do not see a broad quantitative effectiveness evaluation that supports the paper’s soft-search claims. Similarly, the theory still appears fundamentally assumption-driven and conditional. The hypothesis is an assumption that the authors are reluctant to frame as such. Finally, the proposed writing and structural revisions sound promising, but cannot be sufficiently evaluated at the current stage.

**Key Questions For Authors:**

See previous section.

**Limitations:**

The paper uses the generic impact statement. See previous section for limitations and suggestions for improvement.

**Strengths And Weaknesses:**

The paper addresses a timely and important problem. Search algorithms are relevant to a number of growing applications, such as training-data auditing, contamination analysis, and investigating model behavior. The problem is especially important as corpora continue to grow in size, having reached trillion-token scales and expanding. The numerical results show a clear advantage for the proposed algorithm over extant methods, particularly in online search speed.

Despite these strengths, the paper contains major weaknesses that undermine its proposed contributions and novelty, which must be addressed before being considered for publication. Below is a list of those weaknesses in no particular order of importance.

1. Evaluation and Rigor

A main weakness of the paper is related to its biggest claimed strength. The numerical results show significant gains over prior methods, but they do not tell the full story. First, the millisecond results in Figures 3, 4, and 8 refer to isolated \emph{online} queries over a heavily preprocessed index and do not reflect end-to-end search cost. The proposed system requires substantial offline indexing, and while the paper does report index construction time and disk usage, these costs are not grounded proportionally to their practical importance.

The proposed algorithm does show favorable offline preprocessing time relative to some baselines, but this comes with a very large storage footprint. More importantly, the paper does not sufficiently disentangle algorithmic improvement from the underlying time–memory tradeoff. While the proposed system is faster online, it also relies on substantial offline indexing and a very large disk-resident structure. The paper should be much more explicit about this tradeoff and justify why it is favorable in realistic deployments.

The paper also notes that some competing methods, such as SoftMatcha, exceed RAM limits at the largest scales considered. However, it remains unclear whether the proposed method is genuinely less memory-intensive overall, or whether it primarily shifts resource requirements from RAM to disk. The evaluation does not provide a clean RAM vs disk vs total-memory comparison across methods, which makes it difficult to assess practical efficiency.

Second, the empirical evaluation remains too narrow. The paper primarily emphasizes latency-based metrics, especially p95, while omitting a fuller runtime profile such as mean latency and clearer reporting of peak RAM usage and total memory footprint. More importantly, for a paper with a main contribution of semantic soft search, it does not provide a broad quantitative evaluation of retrieval quality or relevance. Prior work such as SoftMatcha reports effectiveness-oriented metrics, whereas this paper largely limits itself to runtime, storage, and qualitative examples.

2. Presentation and Motivation

The paper is poorly structured and not sufficiently self-contained. The introduction begins with a broad and lengthy discussion of recent progress in Large Language Models (LLM) and eventually arrives at search-related applications. It then spends considerable space emphasizing the growth of corpus size, but without clearly articulating the technical challenges this creates for the problem at hand. The paper eventually states what its algorithm does, but it does not sufficiently unpack the underlying bottlenecks, the failure modes of existing approaches, or why its particular combination of techniques is the right response. The more concrete challenge statement appears only later in Section 3, where the paper identifies the two main challenges as slow naive suffix-array lookup and exponential candidate growth. This makes the introduction feel under-motivated relative to what the method actually hinges on.

The related-work sections (2.1 and 2.2) are mostly framing and positioning. The paper spends substantial space on broad background in corpus linguistics, string matching, information retrieval, and dense retrieval before arriving at the most direct comparisons in Section 2.3. As a result, these sections read mostly like an extended introduction rather than a sharply comparative related-work discussion. Even in Section 2.3, where the most relevant baselines are discussed, the paper often states high-level distinctions rather than carefully isolating what is genuinely new algorithmically.

Most importantly, the main body is a high-level description of the method, and many core choices are presented more as engineering decisions than as fully justified technical responses to the problem. The main algorithm is explicitly labeled "simplified", the staged suffix array is deferred to Appendix H, the additional pruning techniques to Appendix I, and the theory details to Appendix J. Thus, while the main body introduces the claimed contributions, many of the details needed to understand or assess them are deferred to the appendices. Given the page limit, it would have been more appropriate to devote more of the main paper to the proposed technical contributions rather than to broad background material.

The paper also assumes too much prior knowledge from the reader. The full setup should be defined more explicitly and self-contained in the main body. Instead, many concepts are introduced with only brief citation-level justification. An expert reader can perhaps follow the intended meaning, but the burden is clearly placed on the reader.

Lastly, the notation is introduced too loosely. Several symbols appear without clear definition, forcing the reader to infer their meaning from context. For example, $\mathcal{V}$ is used for the first time on page 5, Section 4, but never defined. The paper would benefit from a more rigorous and self-contained presentation of the problem setting and notation.

3. Disconnect Between Motivated Contributions and Actual Contributions.

There is a substantial disconnect between what the paper motivates and what it actually establishes through theory and experiments. For example, at the end of page 1 the paper identifies three "unique needs of modern corpus searching" and then states, "We propose a fast full-text search algorithm that meets the three desiderata (1–3) mentioned above". However, only the first point, namely scale, is directly and convincingly supported by the paper’s results. The other two are asserted as motivation, but not meaningfully established by the theory or experiments. In particular, the paper does not directly show that the method improves LLM auditing workflows, and the motivation around non-deterministic training and next-token prediction is stated much more strongly than it is demonstrated. These claims should therefore either be substantially toned down or supported with direct evidence. The paper would be stronger if it focused more narrowly on what it actually addresses, rather than appealing to broader connections that it does not substantiate.

Conversely, the paper’s main technical contributions are not well motivated in the introduction, nor are they sufficiently developed through the related-work discussion in Section 2. For example, why is a suffix-array-based approach more appropriate here? Why is enumerating candidate patterns and pruning them the right design choice? Why are these challenges difficult, and why are existing methods insufficient? What is the role of the disk-aware staged suffix array, and why does it address the problem better than a more modest extension (for example, by moving from RAM to disk) of prior work? The paper later achieves a large run-time disk compression that is never mentioned in the introduction. Similarly, the advantage of soft search vs exact search (in finding more results) is numerically showed but not sufficiently motivated. Even the summarized contributions stated at the end of the introduction are not discussed prior to their summary!

These are central questions for understanding the paper’s actual contribution.

4. Novelty and Contributions

The paper’s novelty appears to lie primarily in a nontrivial engineering combination of existing ideas, rather than in a fundamentally new algorithmic principle. In particular, the proposed fast exact lookup component seems to rely heavily on the fact that the target queries are short: Appendix H states that the method exploits short queries, and uses $L=12$ in the experiments. The resulting two-stage lookup strategy stores a sampled index in RAM, uses it to identify a small candidate block, and then searches only that block on disk. The paper itself notes that this design is similar to Google's BigTable. Thus, the contribution is more naturally framed as an external-memory engineering procedure tailored to trillion-token, short-query corpus search rather than a fundamentally new "disk-aware" approach as motivated.

The proposed method also enumerates query-similar candidate patterns, repeatedly prunes them, and verifies survivors through exact lookup. This makes the effectiveness of pruning central to the method’s practicality, and raises a potential limitation when pruning is less effective than in the corpora studied here. The paper argues that this combinatorial growth is mitigated, but the justification is only partial.

The clearest demonstrated contribution is faster online search over heavily preprocessed data. Although the paper also includes a contamination-detection application, it remains unclear how much this speedup changes a realistic downstream workflow, for example in LLM auditing (as the paper motivates). In that sense, the practical contribution is not fully clear beyond improved online speed. The contamination contribution shows usefulness, but not enough to rescue the broader novelty claim.

Many of the paper’s design choices are only stated, and the reasoning behind those choices is not well articulated. This seems ad-hoc. For example, the paper introduces a modified similarity function and states that it improves over the plain minimum, but provides little analytical or empirical justification for that claim. Similarly, choices such as $m=5$ or $L=12$, and the adaptive-thresholding target $K$ are introduced with limited justification.

The theoretical analysis in Section 4 is very weak. Hypothesis 1 is more of an assumption than a validated hypothesis, and the paper justifies it only by stating that it is believed to hold "from a linguistic perspective". The resulting theory is therefore built on author-introduced assumptions rather than on established properties of the model or corpus. Moreover, the main results are explicitly presented as informal theorems, and the second theorem additionally relies on a Zipf-law assumption for n-gram frequencies. The main sublinearity claim is therefore conditional on multiple strong assumptions.

There also appears to be a mismatch between the theory and the implemented system. The theory analyzes the average number of exact lookups in the simplified Algorithm 1 under fixed thresholds, whereas the practical system uses adaptive thresholding and additional heuristics. As a result, it is unclear how directly the empirical scaling plots correspond to the theoretical claims. The non-monotonic behavior in Figure 8 may not be inconsistent with the stated theory, but it is left unexplained. Taken together, these issues substantially weaken the paper’s claims to novelty, rigor, and well-justified contribution


Minor comments:

The English queries were generated and adjusted using Gemini 3.0 Pro rather than drawn from a natural workload. That weakens the external validity of the latency evaluation.

The appendix explicitly says cache was not cleared after each query, so later queries may benefit from retained cached data. That matters for interpreting the latency numbers, particularly in the millisecond scale.

The paper is really evaluating a heavily pre-indexed deployed system, not raw corpus search. That may be worth stating explicitly once.

The paper claims to "theoretically show" mitigation of exponential growth, which is stronger than what Section 4 justifies. That mismatch is worth pointing out explicitly.

The paper does not clearly separate what is novel relative to the literature (such as SoftMatcha) versus what is novel relative to external-memory indexing ideas more broadly.

---

> ### Author Rebuttal · Authors · 2026-03-31
>
> Thank you for your careful and critical reading. We recognize that insufficient prominence of key discussions in our main text, rather than missing content itself, underlies several of the concerns raised. To respect the character limit, we have prioritized addressing the most significant points here, while ensuring all your suggestions will be reflected in the final version of the manuscript.
> ## Design Choice
> - Since sub-second interactivity is unattainable on massive corpora without an index, pre-indexing is the standard for high-performance search. Indeed, all baseline methods, as well as ours, rely on prior indexing. We argue that our method is not only faster at runtime, but also offers better indexing than infini-gram, the only baseline method scalable to the full FineWeb-Edu corpus, in realistic deployments.
> - Regarding indexing time, as reported in Section 5.4, our method is faster than infini-gram (53.8h vs. 61.2h). Although this indexing time may seem long, it is required only once and would not be a major problem in practice.
> - Regarding index size, our total index is 21.6 TB, but in fact only 5.6 TB is accessed frequently and needs to reside on SSDs (Appendix E). The remaining 16 TB requires only about 10–20 random accesses per query, which can be practically offloaded to HDDs with negligible additional latency (~0.1–0.2s). In contrast, infini-gram uses a plain suffix array and requires its entire 9.9 TB index to reside on SSDs.
> - It is true that our method shifts resource requirements from RAM to disk by loading some part of the index from disk at runtime. Although it is possible to perform a similar shift is possible with SoftMatcha, its latency is high because its algorithm is not disk-aware (discussed in Section 5.3).
> ## Design Choice (Hyperparameters)
> - Regarding query length L=12, queries are known to be very short in most use cases. Macklovitch et al. (2008) report that 97.8% of search queries are 5 words or fewer. In our preliminary user study, more than 99% of queries were shorter than 11 words.
> - Regarding the number of initial search results (K=20), in general retrieval/search systems such as Google Search, users expect the top results to be returned quickly. We followed this common practice.
> - Regarding the softmin (smooth minimum) function and $β$, it balances two conflicting goals: focusing on the most deviating words while still accounting for the others. Plain minimum (as in SoftMatcha) ignores non-minimum similarities, e.g. “gold medal” and “gold medals” score identically against “silver medal”. Plain summation over-penalizes multiple mismatches, e.g. “Theorem 1” scores closer to “Player 1” than to “Lemma 2”. Our smooth minimum interpolates between these extremes via the parameter $β$.
> ## Novelty and Contributions
> Although Google’s BigTable employs a similar two-staged indexing scheme, it is not based on suffix arrays and is similar to ours only in high-level approach, not in the concrete algorithm or data structure. The high efficiency of our exact lookup component is achieved by exploiting assumptions specific to corpus search (e.g. moderate query length). As presented in Section 3.3, we combined this exact lookup component with dynamic corpus-aware pruning in a clean way. This overall algorithmic design tailored to corpus search is one of our main contributions.
>
> To support our theoretical analysis, we empirically verified that Hypothesis 1 holds on FineWeb-Edu. Please see our response to Reviewer vbqW.
>
> ## Coverage of desiderata (2) and (3)
> We appreciate this observation; this feedback is a concrete way to improve the paper! We agree that our presentation does not make it sufficiently clear where and how each desideratum is addressed. We will restructure the introduction to explicitly connect each desideratum to its corresponding technical choice and experiment.
>
> Regarding “Desideratum (2): Searching for similar text”, the paper provides substantial evidence across multiple sections: the qualitative evaluation (§5.1, Table 4) shows retrieval of semantically similar patterns with substitution, insertion, and spelling variation; and the multilingual evaluation (Appendix C, Table 5–9) demonstrates the same across five additional languages; and contamination detection (§6, Tables 14–17) provides 36 concrete cases found only by soft search. We acknowledge that this evidence is dispersed, making it easy to miss.
>
> Regarding “Desideratum (3): Respecting word order”, this property is guaranteed by construction through suffix arrays, which index contiguous token subsequences in their original order.  Lookup succeeds only when tokens appear consecutively in the correct order, unlike bag-of-words–based methods (TF-IDF, BM25) or dense vector search.  This guarantee is one of the reasons why we chose suffix arrays.
>
> We are committed to making these connections and rationales explicit throughout the revised paper.

---

> > ### Author Rebuttal · Reviewer_ht7D · 2026-04-01
> >
> > Thank you for the thoughtful rebuttal. I appreciate the additional clarification on the indexing tradeoff, the rationale behind several hyperparameters. That said, my main concerns remain only partially addressed.
> >
> > First, the most important issue for me is still the narrow empirical evaluation. The rebuttal gives a better explanation of why the indexing tradeoff may be favorable in practice, but it does not address my core concern that the paper mainly demonstrates speed, not semantic search quality. Pointing again to qualitative examples and contamination cases is helpful, but it is not a substitute for a broader quantitative evaluation of retrieval effectiveness or relevance.
> >
> > Second, the theoretical concerns remain. While Hypothesis 1 was empirically checked in the rebuttal, the theory still appears conditional on its strong assumptions, explicitly informal, and not tightly aligned with the implemented system. I maintain that this is not a hypothesis and an assumption that should be explicitly stated. The mismatch between the fixed-threshold analysis and the adaptive-threshold practical system does not seem resolved.
> >
> > Third, several of my presentation concerns are not really answered. For a technically ambitious paper, structure and self-contained exposition are part of the evaluation. My concern was not only that certain points were easy to miss, but that many central technical details and justifications are still too appendix-dependent and insufficiently developed in the main body.
> >
> > Relatedly, the current writing and structure make it difficult for me to judge how fully these issues would actually be resolved in a revision. Many of the rebuttal responses take the form of promises to restructure, clarify, or make connections more explicit. While those revisions may well improve the paper, I cannot determine from the rebuttal alone whether they would adequately address the underlying problems of organization, self-containedness, and technical exposition. I also understand that it is not possible to revise the paper per ICML procedure, but the core writing issue is beyond a promise to revise in my humble opinion.
> >
> > Fourth, on the broader motivation, I appreciate the clarification of how desiderata (2) and (3) are meant to be supported. This helps, but I still think the paper’s framing remains broader than what is directly established. The evidence supports scalable order-sensitive soft corpus search and a contamination-detection use case. It still does not clearly establish the stronger claims around LLM auditing workflows or the broader motivational connection to LLM training.
> >
> > There are also a few points from my review that were not addressed here, including the use of Gemini-generated queries, the cache protocol, and the unexplained non-monotonic behavior in Figure 8.
> >
> > I continue to view the paper as having a real systems contribution and strong runtime results, but with insufficient evaluation breadth, weak and only loosely connected theory, and presentation issues significant enough to limit its current readiness. For these reasons, my overall evaluation remains unchanged.

---

> > > ### Author Response · Authors · 2026-04-08
> > >
> > > Thank you for constructive and critical comments and for further clarifying your concerns. Within the word limit, we focus on as many points as possible, prioritizing the ones you emphasized.
> > > ## Empirical Evaluation
> > > Regarding your earlier comment
> > >
> > > > Prior work such as SoftMatcha reports effectiveness-oriented metrics, whereas (...)
> > >
> > > we have posted a reproduction of the SoftMatcha paper's experiments in our reply to Reviewer tgP5, which confirms that our method works better than SoftMatcha. Sorry for not linking it here.
> > >
> > > To further add tasks where both word order and semantic similarity matter, we also plan to include tasks similar to paraphrase detection. Specifically, we test whether methods can correctly retrieve sentences that are edited versions of a natural text (different on the surface but semantically close to the original). In a preliminary experiment with 400 texts, our method outperformed prior works, infini-gram and SoftMatcha. We will add the full results in the final version.
> > >
> > > ||infini-gram|SoftMatcha|Ours|
> > > |-|-:|-:|-:|
> > > |R@1|65.5|74.5|**84.3**|
> > > ## Theory
> > > **Fixed vs. Adaptive Thresholds** We measured |L_i|/|S_i| on FineWeb-Edu (8.6B tokens) at various and adaptive thresholds $\alpha$. The geometric decay rate $r$ of |L_i|/|S_i| per step (from i=3 to 5) is 0.508x, 0.321x, 0.208x, 0.112x for fixed α=0.8, 0.7, 0.6, 0.5, and 0.111x for adaptive. Since $r$ is well below 1 at every threshold, Hypothesis 1’ (b) holds regardless of $\alpha$, including the adaptive one. Since (1) the adaptive threshold progressively relaxes α and (2) our k-gram and last-bits prunings strictly reduce the number of lookups, the theoretical upper bound derived under fixed thresholds remain valid for the implemented system.
> > >
> > > **Terminology** The “Informal” labels on Theorems 1 and 2 indicate simplified statements; we will move the formal versions from Appendix J to the main text. We also agree with your feedback and will rename “Hypothesis 1” to an explicitly stated “Assumption.”
> > > ## Writing
> > > Our concrete revision plan is as follows:
> > >
> > > - **Related work distant from our main contribution**: We will move the related work section, especially 2.1 and 2.2, to the Appendix.
> > > - **Concrete techniques**: Using the freed space, we will move the algorithmic and theoretical details from Appendix H--J with Fig. 7 back into the main text, at least the main statements and key idea of each design (e.g., reducing random disk accesses and searched patterns). We reflected on uncritically following the recent ML trend of keeping only a sketch in the main text and pushing details to the appendix; you are right it can be made much more self-contained.
> > > - **Motivation-solution link**: To clarify the motivation-solution link for (1) scalability, (2) text similarity, and (3) word order: **(a)** On technical choices, we will state at the start of Sec. 3 that suffix arrays are essentially the only viable option for (1) and (3), and word embeddings under the distributional hypothesis for (2), unlike alternatives such as inverted index or edit distance. **(b)** On motivation for (2) and (3), we will ground it in real use cases, e.g. contamination detection and information retrieval, where natural queries (data with word order and meaning) are the actual search targets.
> > > ## Gemini-generated Queries
> > > To make the evaluation more reliable, we conducted additional latency experiments with real-data as follows. The results are within the same order of magnitude as the original queries, suggesting that our latency evaluation is robust to the choice of query source.
> > >
> > > |(ms)|median|p95|
> > > |-|-:|-:|
> > > |ORCAS (Bing-related search queries)|54.60|433.50|
> > > |FineWeb-Edu substrings|69.97|407.65|
> > > |Gemini (original)|66.35|187.30|
> > >
> > > ## Non-monotonicity in Fig. 8
> > > Thank you for the insightful observation. We re-ran the experiment with a larger sample size and found that the non-monotonicity was an artifact of small samples; with more queries, the results align with the theory. Specifically, p99 with n=400 is determined by only 4 extreme outliers, so we rescaled to n=2000. The resulting p99 latencies on full prunings (Fig. 8 blue line) are as follows. Note that since p99 is only used in Fig. 8, it does not affect overall results.
> > >
> > > |(ms)|17M|87M|436M|2.2B|11B|54B|273B|
> > > |-|-:|-:|-:|-:|-:|-:|-:|
> > > |p99 latency|186.08|253.83|287.38|324.63|340.23|372.40|391.20|
> > > ## Cache Protocol
> > > Although we initially retained the cache (except for 1st run) to simulate the demo tool, we agree on adding cold-cache measurement for complete evaluation. We measured p95 latencies (on FineWeb 8.6B) with the cache cleared after each run. As expected, the latencies consistently increased without the cache, while the relative ranking among methods was preserved---so cache protocol does not materially affect the conclusions.
> > >
> > > |(ms)|w/ Cache|w/o Cache|
> > > |-|-:|-:|
> > > |Ours|187.30|252.84|
> > > |SoftMatcha|28,672.56|35,551.02|
> > > |Ours/exact|0.35|0.87|
> > > |infini-gram/exact|8.31|8.65|
> > > |infini-gram mini/exact|32.89|238.98|

---

### Official Review · Reviewer_vbqW · 2026-03-12

**Soundness:** 3
**Presentation:** 3
**Significance:** 3
**Originality:** 3
**Overall Recommendation:** 4
**Confidence:** 4

**Summary:**

The paper addresses the soft (semantic) pattern matching on a trillion scale corpus. The similarity of a query and a candidate text span is measured by some soft version of minimum cosine distance between individual word embeddings. The method proposed is based on suffix array with two key algorithmic ideas: fast exact lookup enabled by a disk-aware design, and dynamic corpus-aware pruning. Theorectical results were provided, exploiting the Zipf distribution typical in natural language text corpus. Experimental results show that the proposed method obtains speed up up to 33x.

**Compliance With Llm Reviewing Policy:**

Affirmed.

**Final Justification:**

I thank the authors for the constructive rebuttal. The additional experiments address my main concerns and strengthen the paper’s soundness.

The work is original and practically useful, with strong system-level contributions. However, it is better viewed as lexical-semantic matching, with some limitations in compositional semantics, which somewhat limits its broader impact.

Overall, the rebuttal reinforces my prior assessment. I maintain my weak accept recommendation and encourage clearer scope definition in the final version.

**Key Questions For Authors:**

1. Precisions are reported in the contamination experiments, what about recalls?
2. Ablation studies are need for some of the important hyper-parameters, such as 𝛽
3. Which algorithm did you use in Faiss? IVF or HNSW?

**Limitations:**

yes

**Strengths And Weaknesses:**

S1: Nice original engineering achievement with impressive performance speedup.

S2: The pruning is based on a clever observation of the Zipf law, with clear justification (Sec 4), resulting in the sublinear scaling results.

S3: The paper evaluated 6 languages with many metrics, demonstrating the broader application of this method.

W1: The method works at the word level, hence cannot deal with compound words or phrases (e.g., Hong Kong). In addition, the use of suffix array means the method is sensitive to word/phrase ordering, which may arise frequently in practice. Further more, "importance of the machine learning" may not match "significance of machine learning algorithms" do the large norm of "algorithms".

As such, the proposed method is better described as lexical-semantic matching, rather than semantic matching.

W2: The theoretical derivation hinges upon Hypothesis 1, which was testable but not done. This weakens the validity of Thms 1 and 2. Specifically, the Hypothesis assumes query-similar patterns have the same corpus occurrence rate as random patterns. But query-similar patterns are semantically coherent and thus more likely to occur than random 𝑖-grams.

---

> ### Author Rebuttal · Authors · 2026-03-31
>
> ## Theoretical analysis
>
> We appreciate the reviewer's suggestion to empirically verify Hypothesis 1. This prompted a thorough investigation that not only directly confirmed Theorem 1 but also yielded a more precise formulation.
>
> We measured $|L_i|$, $|S_i|$, $|C^*_i|$ on FineWeb-Edu (8.6B tokens, $|V|$=400K, $α$=0.55) and computed their log-shifted geometric mean. Without pruning, $|S_i|$ grows up to 4–5$\times$ per token, reaching 11K+ at i=5. With pruning, $|L_i|$ peaks at i=2 (~50.2), then decreases. This agrees with Theorem 1.
>
> Next, examining $|L_i|$,  $|S_i|$ exhibited the following: (a) At small i, $\frac{|L_i|}{|S_i|}$ is almost the same as $\frac{|C^\*_i|}{|V|^i}$ (e.g., $\frac{|L_1|}{|S_1|}$=1.002, $\frac{|C^\*_1|}{|V|}$=0.957), backing up the original hypothesis; (b) For larger i, $\frac{|C^\*_i|}{|V|^i}$ collapses due to corpus-size saturation against $|V|^i$, but $\frac{|L_i|}{|S_i|}$ continues decreasing geometrically at a rate of 0.16x/step. This reflects accumulated grammatical and collocational constraints: $S_i$ generated by independent word substitution increasingly violates cross-position linguistic restrictions as length grows, which is a well-known natural-language property (Chen & Goodman 1999).
>
> Based on these observations, we will refine Hypothesis 1 as follows.
>
> Hypothesis 1’: (a) when $\frac{|C^\*_i|}{|V|^i}$ is non-negligible, $E[|L_i|]=O(E[|S_i|] \frac{|C^\*_i|}{|V|^i})$; (b) otherwise, $E[\frac{|L_i|}{|S_i|}] = O(r^i)$ for some 0<r<1.
>
> We confirmed that Hypothesis 1’ holds empirically (as shown above) and is sufficient to prove Theorems 1 and 2 without modification. We will revise the paper in the revised version.
>
> ## Ablation of Hyperparameters
>
> We thank the reviewer for this suggestion. We conducted experiments with $β \in \\{1.001,10,10^4,10^{18}\\}$ on 8 representative queries across two corpus sizes. Our similarity function generalizes plain minimum ($β → \infty$, used in SoftMatcha) and summation ($β → 1$). Here, we explain why both extremes are problematic and how $β=10^4$ balances them.
>
> $β = 10^{18}$ (plain minimum): Only the worst-matching word determines similarity; all other positions are ignored. For “olympics gold medalist”, “olympic gold medallist” (85.9%) and “olympic silver medallist” (84.0%) receive nearly identical scores despite very different semantic changes. For “gold medal”, “silver medals” scores 83.3%, nearly matching the exact hit. For “homemade bombs”, results are dominated by patterns like “bombs exploded” (56.1%) and “bombs detonated” (56.0%) where the first word “homemade” is entirely lost; the minimum is determined solely by the second word.
>
> $β = 1.001$ (sum-like): Multiple small deviations are cumulatively penalized. For “theorem 1”, “Player 1” would receive a high score because the per-word cost is summed. For “shortest flight route”, “longest airline routes” and “shorter aircraft routes” disappear entirely from the top-20, losing useful semantic diversity.
>
> $β = 10^4$ (our default): For “olympics gold medalist”, “olympic gold medallist” receives 82.0%, appropriately below the exact match but above genuinely different patterns. For “homemade bombs”, results correctly include “home-made bomb” (57.2%) and “handmade bomb” (52.9%), all semantically relevant while excluding unrelated patterns that appear at extreme β values. For “stock price change”, the top-20 includes “market price change” (67.4%), “share price change” (61.0%), and inflectional variants, providing a coherent set.
>
> We also quantified this with a pairwise LLM-as-judge evaluation (GPT-5.4) on 279 queries, comparing our softmin ($β=10^4$) against plain minimum (same as SoftMatcha). In both orderings, our method wins decisively: 191–72 and 222–40. This confirms that $β = 10^4$ produces qualitatively and quantitatively superior search results.
>
> Results are largely stable across $β \in [10,10^4]$; behavior changes meaningfully only at the extremes. We will include this ablation in the camera-ready version.
> ### On other hyperparameters
>
> We set $α = 0.45$ so that approximately 90% of queries return the full K=20 results (as stated in §5: only 38/400 queries returned fewer). Lower thresholds produce unrelated matches, e.g., “natural language processing” vs. “natural language software” at 44.6% is already marginal.
>
> $γ = m γ’$ is calibrated (§3.1) so that inserting the 50th-lowest-norm word costs $\frac{1}{e}$ when m=5, ensuring low-information words (“the”, “of”) are penalized lightly.
> ### Recall in the contamination experiments
> Although we are also curious, computing recall requires knowing all true contamination in a 1.4T-token corpus, which we currently do not know how to achieve. However, we would like to note that soft search strictly subsumes exact match: all 338 exact-match detections are also found, plus 36 additional (29 verified true).

---

> > ### Author Rebuttal · Reviewer_vbqW · 2026-04-03
> >
> > Thank the authors for the supplementary results. My questions have been addressed and I'll keep my positive score.

---

> > > ### Author Response · Authors · 2026-04-08
> > >
> > > Thank you again for your encouraging and constructive comments. We are glad that our responses addressed your questions, and we believe the additional experiments motivated by your feedback have substantially strengthened our work. We are also grateful for your suggestion to characterize our method as “lexical-semantic matching,” which more precisely positions our contribution. We will gladly adopt this wording in the final version.
> > >
> > > Given that your score remains *weak* accept despite your positive assessment, we would be grateful to know whether there are any remaining concerns that prevent a stronger overall recommendation. Although we have no further opportunity to respond, we will address your concerns in the final version as much as possible. If you feel that our response has fully addressed your concerns, we would appreciate it if this could be reflected in your overall assessment.
> > >
> > > Finally, regarding one question from your original review that we could not fit within the previous character limit: we do not use **FAISS** or any ANN method (which inevitably introduces approximation errors) for enumerating similar words; instead, we perform on-the-fly exact cosine similarity over the full vocabulary using a closed-form, SIMD-accelerated kernel. For our setting (vocabulary 400,000, dimension 300), this is sufficiently fast. On an AWS EC2 i4i.32xlarge instance (64 cores, 128 threads), it takes only 35.60 ms on average. We will elaborate on this in the final version.

---

### Official Review · Reviewer_tgP5 · 2026-03-15

**Soundness:** 4
**Presentation:** 4
**Significance:** 3
**Originality:** 4
**Overall Recommendation:** 5
**Confidence:** 5

**Summary:**

This paper proposes a fast soft search engine for trillion-scale text corpora. The core idea is to keep the scalability of suffix-array-style exact lookup, but extend it to semantic matching with substitution, insertion, and deletion by combining a disk-aware staged suffix array with dynamic corpus-aware pruning. The pruning is the key trick: instead of exploding over all relaxed query variants, it incrementally filters candidates using whether similar prefixes actually occur in the corpus. On FineWeb-Edu at 1.4T tokens, they report 0.34 ms p95 exact lookup and 278 ms p95 soft search, substantially faster than infini-gram, infini-gram mini, and SoftMatcha at large scale, while also supporting softer matching that the exact methods cannot do. They also show a practical use case in benchmark contamination detection, where soft search finds additional contamination cases missed by exact matching, including semantic variants and templated numeric substitutions.

**Compliance With Llm Reviewing Policy:**

Affirmed.

**Key Questions For Authors:**

1. Right now the paper is very convincing on speed, but much less so on retrieval quality. Can you provide a more systematic search-quality evaluation beyond the contamination case study?

2. How much of the practical gain depends on the specific operating point (similarity threshold, top-K, query length)?

**Limitations:**

yes

**Strengths And Weaknesses:**

Strengths

1. The paper has a very clear systems contribution. It is not trying to learn a new retriever; it is making soft phrase-level search actually practical at trillion-token scale, which is a real gap in current corpus search tools.

2. The core algorithmic idea is pretty nice. Combining a staged suffix-array design with dynamic corpus-aware pruning is a clean way to keep the scalability of exact lookup while avoiding the usual combinatorial blowup of relaxed search.

3. I like that the paper is not just heuristic. It gives a theoretical story for why pruning should tame the search space under natural-language statistics, which makes the method feel more principled than “we tried some pruning and it worked.”

4. Running soft search on FineWeb-Edu at 1.4T tokens with sub-second p95 latency is a meaningful result, and the exact-search latency is also very strong.

5. The paper shows a concrete user case--contamination detection.

Weakness

1. The contamination application is a relatively simple user case where the targeted documents is defined to be strictly similar to the query. The application to tasks where more implicit semantic similarity or reasoning-intensive search might not benefit much from this method.

2. The deployment cost is still pretty heavy -- 21.6TB is a lot for many people to afford using it.

3. The precision of contamination detection is 81%, which is not perfectly precise.

4. The latency might not be as ideal as in the paper's setup when the user wants to take a larger K or lower threshold?

---

> ### Author Rebuttal · Authors · 2026-03-31
>
> ## Empirical evaluation
> Thank you for the insightful questions and comments. Motivated by your feedback, we conducted both qualitative and quantitative evaluations, confirming that our method improves search quality over existing baselines.
>
> ### Qualitative evaluation of search quality.
>
> We compared the quality of search results for FineWeb-Edu (8.6B tokens) for 400 queries used in Section 5 using LLM-as-a-judge. Specifically, we provided GPT-5.4 with the top 20 search results returned by SoftMatcha and our method, and evaluated which method it preferred. To control for ordering bias, we ran the evaluation in the original and reversed orders. The preliminary results are as follows: SoftMatcha: 38.7%; Ours: 61.3% (85-189 and 127-147 for each ordering). Note that we only compared the 274/400 queries for which both models output $K=20$ patterns.
>
> ### Quantitative evaluation on application to information retrieval.
>
> Following the experimental protocol and methodology of Deguchi et al. (SoftMatcha, Appendix F), we extended a standard baseline (BM25) with our method. Specifically, since the original natural language queries are not directly suitable for BM25, we prepared “pattern queries” in this experiment (e.g., “how does the coronavirus respond to changes in the weather” -> [“coronavirus”, “response to weather”, “weather change”, “coronavirus response to weather changes”]), and in soft matching, we also add a “partial scores” to the BM25-score based on the similarities. Preliminary results on standard datasets (TREC-COVID and SciFact) using Precision/Recall@k ($\times 100$) show a gain in retrieval accuracy with our approach.
>
> ||TREC-COVID - P@20|R@1000|SciFact - P@1|R@5|
> |-|-:|-:|-:|-:|
> |BM25|33.8|14.9|**42.7**|58.1|
> |SoftMatcha|35.4|17.7|42.3|60.5|
> |**Ours**|**36.0**|**18.0**|**42.7**|**60.7**|
>
> Both experiments show that the soft insertion and deletion capabilities of our method contribute to improved search performance. We will include more comprehensive results with additional datasets, evaluation metrics, and baselines in the camera-ready version.
>
> ## Index size
> Our total index is 21.6TB, but only 5.6TB is accessed frequently for counting soft-matched sequences, and the remaining 16 TB is accessed only for presenting example instances, which requires only  about 10–20 random accesses per query. One possible system configuration is to store the hot part of the index (5.6TB) on a high-speed NVMe SSD and store the remaining part (16TB) on a RAID0 array of HDDs, which is realistic for a server-class machine. Assuming that the random access latency of the RAID0 array is about 10ms, the additional latency would be only about 0.1-0.2 seconds per query.
>
> ## Hyperparameters
> Latency with larger $K$ or lower threshold:  Increasing $K$ or lowering the threshold increases latency, but in this setting, the increase is usually moderate. For an 8.6B-token subset of FineWeb-Edu, when the threshold floor is set to 0.45, the latency grows less than proportionally with $K$, as follows:
>
> |K|20|40|80|160|320|640|
> |-|-:|-:|-:|-:|-:|-:|
> |median latency|64.78|74.05|79.85|85.82|87.44|93.91|
> |p95 latency|178.28|207.42|277.93|325.70|363.34|497.59|
>
> Latency increases further when the threshold floor is lowered. For example, when it is capped at 0.2, the median latency at $K=640$ rises to 300.10 ms. However, lower thresholds—especially below 0.45 in our experience—also tend to admit more unrelated phrases, which we consider undesirable. For instance, the similarity between “fast searching method” and “start searching using” is already 0.45.
>
> On $L=12$ (maximum query length): Studies on real search workloads show that the vast majority of queries are short: in concordancer logs, 87%+ of queries are $\leq 5$ words (Valli, 2012); in general online search, 99.9% of queries are $\leq 12$ words (Bendersky et al., WSCD 2009). Since index size scales linearly with $L$, increasing $L$ for the remaining <1% of use cases would substantially increase storage cost without significant benefit for most users.
> ## Precision of contamination detection
> On contamination precision (81%): This applies to the 36 additional detections beyond the exact match. Combined with 338 exact-match detections, overall precision is substantially higher. We believe this figure is reasonable for a tool that flags candidates for human inspection.

---

> > ### Author Rebuttal · Reviewer_tgP5 · 2026-04-02
> >
> > Thank the authors for the supplementary results. My questions have been addressed and I'll keep my positive score.

---

> > > ### Author Response · Authors · 2026-04-08
> > >
> > > Thank you again for your encouraging and constructive comments. We are delighted to hear that our previous response addressed all your questions and concerns. We will incorporate your feedback to make the final version more compelling and persuasive.
> > >
> > > We also plan to add further empirical evaluations; if interested, please see our additional response to Reviewer ht7D.

---

### Decision · Program_Chairs · 2026-04-30

**Decision:**

Accept (regular)

**Comment:**

This paper proposes a fast, soft search engine designed for trillion-scale text corpora. The contribution is to maintain the scalability of suffix-array-style exact lookups while extending the functionality to semantic matching, which allows for word substitution, insertion, and deletion. To achieve this, the authors combine a disk-aware staged suffix array with dynamic corpus-aware pruning. As noted by the reviewers, the pruning is very important (as is the case in all 'search' indices).  Rather than exploding over all relaxed query variants, the method incrementally filters candidates based on whether similar prefixes actually occur in the corpus.

The reviewers and I agree that the paper offers a very clear systems-level contribution by making "soft" phrase-level search practical at the trillion-token scale. The method demonstrates sub-second latency for soft searches, which is practically impressive. Additionally, the paper successfully demonstrates a downstream use case in benchmark contamination detection. Reviewers also appreciated that the paper provides a theoretical "story" (under many assumptions which seem to be practically motivated). for an intuition behind the method.

However, there are several weaknesses that the reviewers and I have noted: First, it is not entirely clear to me what the exact formal problem setting is, because the problem definition and the experimental setting seem to be intertwined and introduced together. Some reviewers cited underlying problems of organization, self-containedness, and technical exposition. Relatedly, while the method has very strong empirical runtime results, the presentation could be better formally specified, and  the theory seems only loosely connected to the broader claims. Maybe most importantly, the paper does not discuss (at a technical level) nearest neighbor search indices that solve a related problem of retrieving the closest document in similarity, such as DiskANN, HNSW, or IVF. Even though these methods retrieve without the soft min condition over individual words, they are quite relevant and must be discussed.

The paper also seems to optimize for speed, but much less so on retrieval quality. There are also tradeoffs on memory, and some settings require up to 21.6TB. This seems to be a real bottleneck in practical deployment.

The authors also provided some preliminary LLM-as-a-judge and BM25 quantitative evaluations in the rebuttal which strengthen the experiments. However, a more systematic evaluation of search quality is not present. Despite the significant presentation issues and the need for broader evaluation, the strong runtime results for trillion-scale soft search seem valuable to the community. Given this, I believe a weak accept is appropriate for the paper.